# Stochastic Gaussian Zeroth-Order Optimization: Improved Convergence Analysis under Skewed Hessian Spectra

## Abstract

This paper addresses large-scale finite-sum optimization problems, which are particularly prevalent in the big data era. In the field of zeroth-order optimization, stochastic methods have become essential tools. Natural zeroth-order stochastic methods primarily rely on stochastic gradient descent (SGD). Preprocessing the stochastic gradient using a Gaussian vector defines the method ZO-SGD-Gauss (ZSG), whereas estimating coordinate-wise partial derivatives defines ZO-SGD-Coordinate (ZSC). Compared to ZSC, ZSG often demonstrates superior performance in practice. However, the underlying mechanisms behind this phenomenon remain unclear in the academic community. To the best of our knowledge, our work is the first to theoretically analyze the potential advantages of ZSG compared to ZSC. To facilitate convergence analysis, the quadratic regularity assumption is introduced to generalize the smoothness and strong convexity to the Hessian matrix. This assumption makes it possible to integrate Hessian information into the complexity analysis. We provide a theoretical analysis proving the significant convergence improvement of ZSG. Finally, experiments on both synthetic and real-world datasets validate the effectiveness of our theoretical analysis.

## 1 Introduction

Modern machine learning presents significant challenges for optimization due to the large scale of the problems involved. Contemporary datasets are both enormous and high-dimensional, often with millions of samples and features. Because evaluating the full objective or gradient even once is too slow to be useful, stochastic optimization methods have emerged in response.

Throughout the paper, we aim to solve finite-sum minimization problems of the form

$$\min_{\boldsymbol{x} \in \mathbb{R}^d} f(\boldsymbol{x}) \stackrel{def}{=} \frac{1}{n} \sum_{i=1}^{n} f_i(\boldsymbol{x}). \tag{1}$$

An optimization method that solves the problem 1 with function value access only is known as zeroth-order optimization or black-box optimization (Ghadimi & Lan, 2013; Nesterov & Spokoiny, 2017). In recent years, zeroth-order optimization has attracted widespread attention from both the machine learning community and the optimization community (Nesterov & Spokoiny, 2017; Ilyas et al., 2018). One important application of the zeroth-order optimization is the black-box adversarial attack on deep neural networks (Chen et al., 2017; Zhao et al., 2020; Zhang et al., 2020; Bai et al., 2023). In the black-box adversarial attack, only the inputs and outputs of the neural network are available and back propagation is often prohibited (Papernot et al., 2017). In the above situation, the evaluation of gradient $\nabla f(\boldsymbol{x})$ is infeasible. So, applying zeroth-order optimization methods becomes a natural choice. Additional application scenarios in the field of artificial intelligence where zeroth-order optimization algorithms demonstrate significant effectiveness are deep reinforcement learning (Salimans et al., 2017; Mania et al., 2018; Zhang & Zavlanos, 2023; Jing et al., 2024), hyper-parameter tuning (Snoek et al., 2012; Rapin & Teytaud, 2018), the problem of optimizing functions with only ranking feedback (Tang et al., 2023), learning linear quadratic regulators (Malik et al., 2020; Mohammadi et al., 2020), and so on. Zeroth-order optimization has even played a

significant role in fine-tuning LLMs. Malladi et al. (2023) and Zhao et al. (2024) use the zeroth-order optimization methods for fine-tuning, in addressing the significant memory overhead of first-order optimizers. Zeroth-order optimization achieves a substantial memory reduction and makes it possible to train and store LLMs on low-cost hardware.

The ZO-SGD-Gauss (ZSG) algorithm is based on the Gaussian version of SPSA (Spall, 1992). The ZO-SGD-Coordinate (ZSC) algorithm is based on the finite-difference stochastic approximation (Kiefer & Wolfowitz, 1952). Although the ZSG algorithm and the ZSC algorithm share the same theoretical convergence rate and both have sample complexity that is linear in the dimension (Ghadimi & Lan, 2013), ZSG has a wider range of applications and performs better than ZSC in practice. For example, ZSG has been widely used in fine-tuning LLMs (Malladi et al., 2023; Zhao et al., 2024) and black-box attacks (Ilyas et al., 2018). The academic community is still unclear about the underlying mechanism why ZSG outperforms ZSC. For the gradient descent method, recent works by Yue et al. (2023) and Wang et al. (2024) show that zeroth-order Gaussian gradient descent can outperform coordinate descent under skewed Hessian spectra, which indicates ill-conditioning and anisotropy in the loss landscape. An intriguing question is whether the zeroth-order SGD algorithm possesses a similar property to the zeroth-order gradient descent algorithm. Motivated by these works, we try to investigate whether ZSG can theoretically achieve better complexity than ZSC. We obtain a surprising result: compared to ZSC, ZSG exhibits weak dimensional dependence—meaning that the dimension $d$ does not explicitly appear in the complexity bounds. Our work fills a theoretical gap in the field of zeroth-order optimization.

## 1.1 Literature Review

Here, we present a concise overview of stochastic optimization methods.

An optimization method that solves the problem 1 by accessing gradient information from a subset of samples is called SGD. SGD and its variance reduction variants, which operate on only a small mini-batch of data at each iteration, have become the preferred methods (Robbins & Monro, 1951; Moulines & Bach, 2011; Johnson & Zhang, 2013; Allen-Zhu, 2018). However, stochastic optimizers sacrifice stability in favor of speed. Parameters such as the learning rate are challenging to choose (Nemirovski et al., 2009), and for ill-conditioned large-scale machine learning problems, even finding the optimal learning rate can lead to very slow convergence. Second-order optimizers based on the Hessian, such as Newton's method (Battiti, 1992) and quasi-Newton methods (Dennis & Moré, 1977; Jin & Mokhtari, 2023), are the classic remedy for solving above challenges. Some researchers have proposed using stochastic Hessian approximations while still utilizing the full gradient (Lacotte et al., 2021; Tong et al., 2021). Then, Frangella et al. (2022) propose the SketchySGD algorithm whose excellent performance suggests it could potentially replace SGD.

When the gradient is difficult to calculate or cannot be obtained, researchers shift their attention from the study of SGD to stochastic zeroth-order optimization algorithms, estimating the gradient using function value differences (Ghadimi & Lan, 2013; Duchi et al., 2015; Nesterov & Spokoiny, 2017). Malladi et al. (2023) directly use zeroth-order optimizer (ZOO) for fine-tuning LLMs. However, the zeroth-order optimization algorithms mentioned above overlook the use of higher-order information about the objective, leading to less competitive convergence in practice. Similar to the development of SGD, researchers have begun to introduce second-order Hessian information into zeroth-order optimization algorithms. This idea holds promise for the design of efficient and competitive algorithms. Chen et al. (2017) utilize the second-order Hessian information in a relatively coarsened manner. Ye et al. (2018) take a first step to efficiently incorporate second-order Hessian information of the objective function and propose a novel class of algorithms called the ZOHA algorithm. Zhao et al. (2024) propose HiZOO, which is the first work to leverage the diagonal Hessian to enhance ZOO for fine-tuning LLMs.

It is worth noting that Nesterov & Spokoiny (2017) conduct a theoretical analysis of the complexity bounds for three random gradient-free oracles. However, the potential advantages of using Gaussian preconditioning vectors remain unexplored. The essential reason is that they do not effectively utilize the information from the Hessian matrix in their theoretical analysis process. Therefore, in essence, our work is different from that of (Nesterov & Spokoiny, 2017). In addition, we would like to highlight some differences between previous works and ours. Although Malladi et al. (2023) propose the descent theorem for ZO-SGD, the ultimately proven con-

vergence rate $t = \mathcal{O}\left((r+1) \cdot \left(\frac{L}{\mu} + \frac{L\alpha}{\mu^2 |\mathcal{S}|}\right) \log \frac{f(\boldsymbol{x}^0) - f^*}{\epsilon}\right)$ is essentially that of gradient descent rather than stochastic gradient descent, where $r$ is effective rank and $\alpha$ is used to control the covariance of the gradient estimation. So, Malladi et al. (2023) do not reveal the true convergence rate of ZSG, and determining it remains a challenging open problem. Yue et al. (2023) enhance the convergence rate $\mathcal{O}\left(\frac{\sum_{i=1}^d \lambda_i\left(\nabla^2 f(\boldsymbol{x})\right)}{\mu} \log \frac{1}{\epsilon}\right)$ for standard zeroth-order optimization algorithm (Nesterov & Spokoiny, 2017) on quadratic objectives, where $\lambda_i$ represents the $i$-th eigenvalue of $\nabla^2 f(\boldsymbol{x})$. Similarly, their work is based on gradient descent rather than SGD. The iterative algorithm $\boldsymbol{x}^{t+1} \leftarrow \boldsymbol{x}^t - \alpha\left(\frac{1}{n}\sum_{i=1}^n \text{clip}_C\left(\frac{f_i(\boldsymbol{x}^t + \alpha \boldsymbol{s}_t) - f_i(\boldsymbol{x}^t - \alpha \boldsymbol{s}_t)}{2\alpha}\boldsymbol{s}_t\right) + \boldsymbol{u}_t\right)$ proposed by Zhang et al. (2023) still relies on full gradient information rather than stochastic gradient information. In summary, our theoretical analysis is different from above works. The theoretical result we obtain is unique.

## 1.2 CONTRIBUTIONS

The main contributions are summarized as follows:

- Compared to ZSC, we establish an accelerated convergence rate for ZSG. We successfully reveal that ZSG also exhibits weak dimensional dependence, which explains the fundamental reason behind its superior empirical performance. The advantage of ZSG becomes more pronounced under skewed Hessian spectra. It is worth noting that, in practice, this condition often holds because the singular values of Hessian matrices tend to decrease rapidly (Yue et al., 2023). To the best of our knowledge, our work is the first to theoretically analyze the potential advantages of ZSG compared to ZSC and our conclusion is novel.

- Our theoretical analysis is based on the quadratic regularity assumptions. This assumption helps leverage Hessian structure and broadens the applicability of our complexity results beyond the quadratic case.

- Our research indicates that ZSG also exhibits weak dimensional dependence, similar to zeroth-order gradient descent. This fills a theoretical gap in the field of zeroth-order optimization, and our analytical results provide significant theoretical insights.

- Extensive experiments confirm the reliability of our theoretical analysis. On both synthetic and real-world datasets, the performance of ZSG outperforms that of ZSC. This observation is in line with established practices in the optimization community.

## 2 NOTATION AND ASSUMPTIONS

Let us define the weighted Euclidean norm and weighted inner product associated with a positive definite weight matrix $\mathbf{M} \succ 0$

$$\|\boldsymbol{x}\|_{\mathbf{M}} \stackrel{def}{=} \langle \boldsymbol{x}, \boldsymbol{x} \rangle_{\mathbf{M}}^{\frac{1}{2}}, \quad \langle \boldsymbol{x}, \boldsymbol{y} \rangle_{\mathbf{M}} \stackrel{def}{=} \langle \mathbf{M}\boldsymbol{x}, \boldsymbol{y} \rangle.$$

We define the stochastic gradient $\nabla f(\boldsymbol{x}, \mathcal{S}) = \frac{1}{|\mathcal{S}|} \sum_{j \in \mathcal{S}} \nabla f_j(\boldsymbol{x})$, where $\mathcal{S}$ represents the sample set and $|\mathcal{S}|$ represents the sample size.

A widely accepted notion is that the assumptions of $f$ being $L$-smooth and $\mu$-strongly convex are standard in the analysis of stochastic gradient methods for solving the problem 1. As research on stochastic algorithms deepens, many researchers have proposed more generalized assumptions. Hanzely et al. (2018) introduce the $\mathbf{M}$-smoothness assumption, which is a common assumption in modern analyses of stochastic methods. Gower et al. (2019) present the relative smoothness assumption and relative convexity assumption to exploit information from the Hessian matrix. Frangella et al. (2023) utilize the quadratic regularity assumption to overcome the dilemma of infrequent preconditioner updates. Frangella et al. (2022) propose the relative quadratic regularity assumption, which replaces the Hessian matrix with any positive definite matrix.

Based on these developments, we present the following assumptions on the objective function $f$. First, we introduce the quadratic regularity assumption (Frangella et al., 2023), which generalizes classical notions of smoothness and strong convexity to the Hessian norm, thereby enabling the incorporation of rich Hessian information into the complexity analysis.

---

**Algorithm 1** ZSG: ZO-SGD-Gauss Method

---

**Input and Initialize:** parameters $\boldsymbol{x} \in \mathbb{R}^d$, loss function $f : \mathbb{R}^d \to \mathbb{R}$, step budget $t$, step size $\eta_t > 0$, perturbation scale $\alpha$, sample distribution $\mathcal{D}$, initial point $\boldsymbol{x}^0 \in \mathbb{R}^d$
**for** $t = 0, 1, \cdots$ **do**
    Sample $\mathcal{S}_t \sim \mathcal{D}$ and $\boldsymbol{u}_t \sim \mathcal{N}(\boldsymbol{0}, \mathbf{I}_d)$
    Query the zeroth-order oracle $f_+^t = f(\boldsymbol{x}^t + \alpha\boldsymbol{u}_t, \mathcal{S}_t)$
    Query the zeroth-order oracle $f_-^t = f(\boldsymbol{x}^t - \alpha\boldsymbol{u}_t, \mathcal{S}_t)$
    Estimating the gradient $\hat{\nabla} f(\boldsymbol{x}^t, \mathcal{S}_t) = \frac{(f_+^t - f_-^t)}{2\alpha} \cdot \boldsymbol{u}_t$
    $\boldsymbol{x}^{t+1} = \boldsymbol{x}^t - \eta_t \hat{\nabla} f(\boldsymbol{x}^t, \mathcal{S}_t)$
**end for**

---

**Assumption 2.1.** Let $f : \mathbb{R}^d \to \mathbb{R}$ be a twice differentiable function, and let $\mathbf{M}$ denote the Hessian matrix of $f$. The function $f$ is said to be upper quadratically regular with respect to $\mathbf{M}$, if there exists a global constant $0 \le \gamma_u < \infty$, such that for all $\boldsymbol{x}, \boldsymbol{y}, \boldsymbol{z} \in \mathbb{R}^d$,

$$f(\boldsymbol{y}) \le f(\boldsymbol{x}) + \langle \nabla f(\boldsymbol{x}), \boldsymbol{y} - \boldsymbol{x} \rangle + \frac{\gamma_u}{2} \|\boldsymbol{y} - \boldsymbol{x}\|_{\mathbf{M}(\boldsymbol{z})}^2. \tag{2}$$

Similarly, $f$ is said to be lower quadratically regular with respect to $\mathbf{M}$, if there exists a global constant $\gamma_l > 0$, such that for all $\boldsymbol{x}, \boldsymbol{y}, \boldsymbol{z} \in \mathbb{R}^d$,

$$f(\boldsymbol{y}) \ge f(\boldsymbol{x}) + \langle \nabla f(\boldsymbol{x}), \boldsymbol{y} - \boldsymbol{x} \rangle + \frac{\gamma_l}{2} \|\boldsymbol{y} - \boldsymbol{x}\|_{\mathbf{M}(\boldsymbol{z})}^2. \tag{3}$$

Next, we introduce the standard variance assumption.

**Assumption 2.2.** The variance of the stochastic gradient can be bounded by $\sigma^2$, which means

$$\mathbb{E}\left[\|\nabla f(\boldsymbol{x}, \mathcal{S}) - \nabla f(\boldsymbol{x})\|^2\right] \le \sigma^2. \tag{4}$$

## 3 ALGORITHM DESCRIPTION

This section commences with a detailed description of ZSG the algorithm. According to the formulation in Nesterov & Spokoiny (2017), the zeroth-order gradient estimator can be expressed as $\hat{\nabla} f(\boldsymbol{x}, \mathcal{S}) = \frac{[f(\boldsymbol{x}+\alpha\boldsymbol{u}, \mathcal{S}) - f(\boldsymbol{x}-\alpha\boldsymbol{u}, \mathcal{S})]}{2\alpha} \cdot \boldsymbol{u}$, where $\boldsymbol{u} \in \mathbb{R}^d$ is sampled from $\mathcal{N}(\boldsymbol{0}, \mathbf{I}_d)$ and $\alpha$ is a very small perturbation scale. In order to help us prove complexity, we need to find the connection between the zeroth-order oracles and the gradient.

**Lemma 3.1.** *We access to the $f(\boldsymbol{x} + \alpha\boldsymbol{u}, \mathcal{S})$ and $f(\boldsymbol{x} - \alpha\boldsymbol{u}, \mathcal{S})$. Through the upper quadratically regular assumption, we yield the following equality*

$$\hat{\nabla} f(\boldsymbol{x}, \mathcal{S}) = \boldsymbol{u}\boldsymbol{u}^\top \nabla f(\boldsymbol{x}, \mathcal{S}) + \phi(\boldsymbol{u}, \alpha, \boldsymbol{x}), \tag{5}$$

*with*

$$\|\phi(\boldsymbol{u}, \alpha, \boldsymbol{x})\| \le \frac{\gamma_u \alpha}{2} \|\boldsymbol{u}\|_{\mathbf{M}(\boldsymbol{z})}^2 \cdot \|\boldsymbol{u}\|, \tag{6}$$

*where $\boldsymbol{z}_1 \in (\boldsymbol{x}, \boldsymbol{x} + \alpha\boldsymbol{u})$, $\boldsymbol{z}_2 \in (\boldsymbol{x} - \alpha\boldsymbol{u}, \boldsymbol{x})$ and $\mathbf{M}(\boldsymbol{z}) = \begin{cases} \mathbf{M}(\boldsymbol{z}_1) & \text{if } \mathbf{M}(\boldsymbol{z}_1) \succeq \mathbf{M}(\boldsymbol{z}_2) \\ \mathbf{M}(\boldsymbol{z}_2) & \text{otherwise} \end{cases}$.*

The detailed proof is presented in C.1. The aforementioned relationships can help us conduct convergence analysis. This paper focuses on analyzing the convergence properties of the following update rule:

$$\boldsymbol{x}^{t+1} = \boldsymbol{x}^t - \eta_t \hat{\nabla} f(\boldsymbol{x}^t, \mathcal{S}_t). \tag{7}$$

The main algorithmic procedure of ZSG is provided in Algorithm 1. We can also obtain the ZSC algorithm by simply substituting $\boldsymbol{u}_t \sim \mathcal{N}(\boldsymbol{0}, \mathbf{I}_d)$ with $\boldsymbol{e}_t \sim \mathcal{U}^d$, where $\mathcal{U}^d$ denotes the uniform distribution over the standard basis vectors in $\mathbb{R}^d$. The main algorithmic procedure of ZSC is provided in Algorithm 2.

## 4 MAIN THEORETICAL RESULTS

This section provides an in-depth examination of the iterative complexity of ZSG under the assumptions we introduced. First, we study the convergence properties of quadratic functions. To explain the superiority of ZSG conveniently, we assume that $f(\boldsymbol{x}) = \frac{1}{2}\boldsymbol{x}^\top \mathbf{M}\boldsymbol{x} - \boldsymbol{b}^\top \boldsymbol{x}$. If the objective function $f$ in Assumption 2.1 is quadratic function, we need to point that $\gamma_l = \gamma_u = 1$ and $\mathbf{M}(\boldsymbol{z}) \equiv \mathbf{M}$, meaning the Hessian matrix is independent of the iteration points.

We begin by presenting several essential lemmas that support the derivation of the main theorems in this section. The detailed proofs of Lemma 4.1 and Lemma 4.2 are provided in Section C. In addition, several other lemmas along with their proofs are given in Section B. The complete proofs of the main theorems and corollaries are deferred to Section D and Section E.

**Lemma 4.1.** *Consider an arbitrary point $\boldsymbol{x} \in \mathbb{R}^d$ and a Gaussian vector $\boldsymbol{u}_t \sim \mathcal{N}(\mathbf{0}, \mathbf{I}_d)$. For any $t>0$, the zeroth-order approximation of the gradient at $\boldsymbol{x}$ admits the following upper bound:*

$$\mathbb{E}_{\boldsymbol{u}_t}\left[\left\|\boldsymbol{u}_t \boldsymbol{u}_t^\top \nabla f(\boldsymbol{x}^t, \mathcal{S}_t)\right\|_{\mathbf{M}}^2\right] \leq 3\mathrm{tr}(\mathbf{M})\left\|\nabla f(\boldsymbol{x}^t, \mathcal{S}_t)\right\|^2. \tag{8}$$

**Lemma 4.2.** *Let $f^*$ denote the optimum of the objective function. For all $t>0$, if $\boldsymbol{z} \in (\boldsymbol{x}^t, \boldsymbol{x}^*)$, the difference between the function value at $\boldsymbol{x}^t$ and the optimum $f^*$ can be bounded as follows:*

$$f(\boldsymbol{x}^t) - f^* \leq \frac{1}{2\gamma_l \lambda_{\min}(\mathbf{M}(\boldsymbol{z}))}\left\|\nabla f(\boldsymbol{x}^t)\right\|^2. \tag{9}$$

**Theorem 4.3.** *Let $f$ be a quadratic function, and assume that f is both upper and lower quadratically regular with respect to $\mathbf{M}$. That is, Assumption 2.1 holds. In addition, the variance of stochastic gradient is bounded, i.e., Assumption 2.2 holds. Suppose the update rule of $\boldsymbol{x}^{t+1}$ follows Eq. 7. We define $P_1(\alpha^2) = \frac{[1+2\lambda_{\max}(\mathbf{M})\eta]\lambda_{\max}^2(\mathbf{M})(6+d)^3\alpha^2}{4\lambda_{\min}(\mathbf{M})}$ and choose $\eta_t \equiv \eta \leq \frac{1}{12\mathrm{tr}(\mathbf{M})}$, then, we obtain*

$$\mathbb{E}\left[f(\boldsymbol{x}^{t+1}) - f^*\right] \leq \frac{6\eta\mathrm{tr}(\mathbf{M})\sigma^2}{\lambda_{\min}(\mathbf{M})} + P_1(\alpha^2) + \left[1 - \frac{1}{2}\eta\lambda_{\min}(\mathbf{M})\right]^t\left[f(x^0) - f^*\right].$$

We can observe that ZSG converges to a ball around the optimum from Theorem 4.3. This phenomenon is analogous to the classic SGD which employs a fixed step size (Moulines & Bach, 2011).

**Corollary 4.4.** *Theorem 4.3 suggests that with a fixed step size, the algorithm may fail to converge in the presence of noise. Assume that f and the parameters satisfy the conditions specified in Theorem 4.3. Since we can choose a sufficiently small $\alpha$ in practice, we can omit it. If $\sigma^2 = 0$, to find an $\varepsilon$-suboptimal solution, the iteration complexity is*

$$t = \mathcal{O}\left(\frac{\mathrm{tr}(\mathbf{M})}{\lambda_{\min}(\mathbf{M})}\log\frac{1}{\varepsilon}\right). \tag{10}$$

When $\sigma^2 = 0$, the update of $\boldsymbol{x}$ depends on the full gradient, reducing to the deterministic setting. The result in Wang et al. (2024) can be viewed as an intermediate product of our analysis. Their purpose is to compare it with the coordinate-sketched SEGA (Hanzely et al., 2018), which achieves an iteration complexity of $\mathcal{O}\left(\frac{d\lambda_{\max}(\mathbf{M})}{\lambda_{\min}(\mathbf{M})}\log\frac{1}{\varepsilon}\right)$ without importance sampling. However, our work focus on the analysis of zeroth-order stochastic optimization. The following theorem and corollary will indicate that ZSG outperforms ZSC.

**Theorem 4.5.** *Let $f$ be a quadratic function, and suppose that Assumption 2.1 and Assumption 2.2 hold. Suppose the update rule of $\boldsymbol{x}^{t+1}$ follows Eq. 7. Assume the step size follows the decreasing form $\eta_t = \frac{l}{\gamma+t}$, where $\gamma > 0$ and the intermediate parameter $l = \frac{3}{\lambda_{\min}(\mathbf{M})}$. Let $t_{\max} = T$ and $\alpha \leq \sqrt{\frac{\alpha_0}{T+1}}$, where $\alpha_0$ is a tunable perturbation scale. We define $Q_1(\alpha_0^2) = \frac{3[6\lambda_{\max}(\mathbf{M})+36\mathrm{tr}(\mathbf{M})]\lambda_{\max}^2(\mathbf{M})(6+d)^3\alpha_0^2}{4\lambda_{\min}^2(\mathbf{M})}$. The initial step size satisfies $\eta_0 = \frac{l}{\gamma} \leq \frac{1}{12\mathrm{tr}(\mathbf{M})}$, which implies $\gamma \geq \frac{36\mathrm{tr}(\mathbf{M})}{\lambda_{\min}(\mathbf{M})}$. Next, define the auxiliary parameter $v = \max\left\{\gamma(f(\boldsymbol{x}^0) - f^*), \frac{54\mathrm{tr}(\mathbf{M})\sigma^2}{\lambda_{\min}^2(\mathbf{M})} + Q_1(\alpha_0^2)\right\}$. Then, for every integer $t$ with $0 \leq t \leq T$, we can obtain the following result*

$$\mathbb{E}\left[f(\boldsymbol{x}^t) - f^*\right] \leq \frac{v}{\gamma+t}.$$

**Corollary 4.6.** *Theorem 4.5 implies that with a decreasing step size, ZSG converges in the presence of noise. Assume that the function f and the parameters satisfy the conditions specified in Theorem 4.5. Then, to obtain an $\varepsilon$-suboptimal solution, the iteration complexity satisfies*

$$t = \mathcal{O}\left(\left[\frac{\text{tr}(\mathbf{M})\sigma^2}{\lambda_{\min}^2(\mathbf{M})} + Q_1(\alpha_0^2)\right]\frac{1}{\varepsilon}\right). \tag{11}$$

When $\sigma^2 > 0$ and a sufficiently small $\alpha_0$ is chosen in practice, the iteration complexity of ZSG is given by $\mathcal{O}\left(\frac{\text{tr}(\mathbf{M})\sigma^2}{\lambda_{\min}^2(\mathbf{M})}\frac{1}{\varepsilon}\right)$. In this case, the zeroth-order oracle is queried twice per iteration. Therefore, the query complexity of ZSG is also $\mathcal{O}\left(\frac{\text{tr}(\mathbf{M})\sigma^2}{\lambda_{\min}^2(\mathbf{M})}\frac{1}{\varepsilon}\right)$. Notably, the iteration complexity of SGD is $\mathcal{O}\left(\frac{\lambda_{\max}(\mathbf{M})\sigma^2}{\lambda_{\min}^2(\mathbf{M})}\frac{1}{\varepsilon}\right)$ (Rakhlin et al., 2011). The prevailing view in the optimization community (Ghadimi & Lan, 2013; Nesterov & Spokoiny, 2017) is that zeroth-order optimization methods typically have $\mathcal{O}(d)$ times the complexity of their first-order counterparts. Thus, the query complexity of ZSC becomes $\mathcal{O}\left(\frac{d\lambda_{\max}(\mathbf{M})\sigma^2}{\lambda_{\min}^2(\mathbf{M})}\frac{1}{\varepsilon}\right)$. To better elucidate our theoretical contribution, we rigorously establish that the ZSC algorithm achieves an iteration complexity of $\mathcal{O}\left(\left[\frac{d\lambda_{\max}(\mathbf{M})\sigma^2}{\lambda_{\min}^2(\mathbf{M})} + C_1(\alpha_0^2)\right]\frac{1}{\varepsilon}\right)$, where $C_1(\alpha_0^2) = \frac{3[6\lambda_{\max}(\mathbf{M})+12d\lambda_{\max}(\mathbf{M})]\lambda_{\max}^2(\mathbf{M})d^2\alpha_0^2}{4\lambda_{\min}^2(\mathbf{M})}$. The core of the proof lies in substituting $\mathbf{u}_t \sim \mathcal{N}(\mathbf{0}, \mathbf{I}_d)$ with $\mathbf{e}_t \sim \mathcal{U}^d$, where $\mathcal{U}^d$ denotes the uniform distribution over the standard basis vectors in $\mathbb{R}^d$. Accordingly, we prove that ZSG enjoys a superior convergence rate over ZSC, and this advantage becomes more pronounced under skewed Hessian spectra.

Next, we generalize our results to other function classes satisfying Assumption 2.1, where $\gamma_l \neq 1$ or $\gamma_u \neq 1$ may hold.

**Theorem 4.7.** *Suppose f is in the general form described in problem 1 and assumption 2.1,2.2 hold. Suppose the update rule of $\mathbf{x}^{t+1}$ follows Eq. 7, using a fixed step size*

$$\eta_t \equiv \eta \le \frac{1}{12\gamma_u\text{tr}(\mathbf{M})}. \tag{12}$$

*We define $\text{tr}(\mathbf{M}) = \max_{\mathbf{z}^t}\text{tr}(\mathbf{M}(\mathbf{z}^t))$, with similar definitions for $\lambda_{\min}(\mathbf{M})$ and $\lambda_{\max}(\mathbf{M})$. We also define $P_2(\alpha^2) = \frac{[1+2\gamma_u\lambda_{\max}(\mathbf{M})\eta]\lambda_{\max}^2(\mathbf{M})(6+d)^3\gamma_u^2\alpha^2}{4\gamma_l\lambda_{\min}(\mathbf{M})}$. Then, it holds that*

$$\mathbb{E}\left[f(\mathbf{x}^{t+1}) - f^*\right] \le \frac{6\eta\gamma_u\text{tr}(\mathbf{M})\sigma^2}{\gamma_l\lambda_{\min}(\mathbf{M})} + P_2(\alpha^2) + \left[1 - \frac{1}{2}\eta\gamma_l\lambda_{\min}(\mathbf{M})\right]^t\left[f(x^0) - f^*\right].$$

*If $\sigma^2 = 0$ and $\alpha$ is sufficiently small, the complexity to achieve an $\varepsilon$-suboptimal solution satisfies*

$$t = \mathcal{O}\left(\frac{\gamma_u\text{tr}(\mathbf{M})}{\gamma_l\lambda_{\min}(\mathbf{M})}\log\frac{1}{\varepsilon}\right). \tag{13}$$

As shown in Theorem 4.7, ZSG under the full-gradient setting achieves a faster convergence rate than the coordinate-sketched SEGA when $\frac{\gamma_u}{\gamma_l} = \mathcal{O}(1)$. This intermediate result extends the result in Wang et al. (2024) from quadratic functions to a broader class of objective functions.

**Theorem 4.8.** *Suppose f is in the general form described in problem 1 and assumption 2.1,2.2 hold. Suppose the update rule of $\mathbf{x}^{t+1}$ follows Eq. 7. Assume the step size follows the decreasing form $\eta_t = \frac{l}{\gamma+t}$, where $\gamma > 0$ and the intermediate parameter $l = \frac{3}{\gamma_l\lambda_{\min}(\mathbf{M})}$. Let $t_{\max} = T$ and $\alpha \le \sqrt{\frac{\alpha_0}{T+1}}$, where $\alpha_0$ is a tunable perturbation scale. We define $Q_2(\alpha_0^2) = \frac{3[6\gamma_u\lambda_{\max}(\mathbf{M})+36\gamma_u\gamma_l\text{tr}(\mathbf{M})]\lambda_{\max}^2(\mathbf{M})(6+d)^3\gamma_u^2\alpha_0^2}{4\gamma_l^2\lambda_{\min}^2(\mathbf{M})}$. The initial step size satisfies $\eta_0 = \frac{l}{\gamma} \le \frac{1}{12\gamma_u\text{tr}(\mathbf{M})}$, which implies $\gamma \ge \frac{36\gamma_u\text{tr}(\mathbf{M})}{\lambda_{\min}(\mathbf{M})}$. Next, define the auxiliary parameter $v = \max\left\{\gamma(f(\mathbf{x}^0) - f^*), \frac{54\gamma_u\text{tr}(\mathbf{M})\sigma^2}{\gamma_l^2\lambda_{\min}^2(\mathbf{M})} + Q_2(\alpha_0^2)\right\}$. We define $\text{tr}(\mathbf{M}) = \max_{\mathbf{z}^t}\text{tr}(\mathbf{M}(\mathbf{z}^t))$, with similar definitions for $\lambda_{\min}(\mathbf{M})$ and $\lambda_{\max}(\mathbf{M})$. Then, for every integer t with $0 \le t \le T$, we can obtain*

$$\mathbb{E}\left[f(\mathbf{x}^t) - f^*\right] \le \frac{v}{\gamma+t}.$$

*Finally, to obtain an $\varepsilon$-suboptimal solution, the iteration complexity satisfies*

$$t = \mathcal{O}\left(\left[\frac{\gamma_u \mathrm{tr}(\mathbf{M})\sigma^2}{\gamma_l^2 \lambda_{\min}^2(\mathbf{M})} + Q_2(\alpha_0^2)\right]\frac{1}{\varepsilon}\right). \tag{14}$$

As shown in Theorem 4.8, when $\sigma^2 > 0$ and a sufficiently small $\alpha_0$ is chosen, the query complexity of ZSG is given by $\mathcal{O}\left(\frac{\gamma_u \mathrm{tr}(\mathbf{M})\sigma^2}{\gamma_l^2 \lambda_{\min}^2(\mathbf{M})}\frac{1}{\varepsilon}\right)$. We rigorously establish that the ZSC algorithm achieves an iteration complexity of $\mathcal{O}\left(\left[\frac{\gamma_u d\lambda_{\max}(\mathbf{M})\sigma^2}{\gamma_l^2 \lambda_{\min}^2(\mathbf{M})} + C_2(\alpha_0^2)\right]\frac{1}{\varepsilon}\right)$, where $C_2(\alpha_0^2) = \frac{3[6\gamma_u \lambda_{\max}(\mathbf{M}) + 12\gamma_u \gamma_l d\lambda_{\max}(\mathbf{M})]\lambda_{\max}^2(\mathbf{M})d^2\gamma_u^2\alpha_0^2}{4\gamma_l^2 \lambda_{\min}^2(\mathbf{M})}$.

The proof of ZSC proceeds along the same lines as the previous analysis. For general functions, we accordingly prove that ZSG enjoys a superior convergence rate over ZSC, and this advantage also becomes more pronounced under skewed Hessian spectra. For quadratic functions, it follows that $\gamma_u = \gamma_l = 1$, which is consistent with our previous analysis in Theorem 4.5. That is, based on assumption 2.1, we establish that ZSG exhibits weak dimensional dependence for both quadratic and general non-quadratic objectives, thereby providing a novel theoretical explanation for the empirical observation that ZSG tends to outperform ZSC in practice.

## 5 EXPERIMENTS

In the preceding sections, we have conducted a comprehensive theoretical analysis of ZSG and ZSC, highlighting their respective convergence behaviors. This section is dedicated to the empirical validation of ZSG's effectiveness and superiority. We choose $\alpha = 10^{-6}$ for all experiments.

### 5.1 QUADRATIC FUNCTIONS

In this part, our experiments will focus on the quadratic minimization problem, whose objective function adheres to the form delineated in the problem 1, characterized by

$$\min_{x\in\mathbb{R}^d} f(\boldsymbol{x}) = \frac{1}{2n}\boldsymbol{x}^\top \mathbf{A}\mathbf{A}^\top \boldsymbol{x} - \boldsymbol{b}^\top \boldsymbol{x},$$

where $\mathbf{M} = \frac{1}{n}\mathbf{A}\mathbf{A}^\top$. The parameters of the quadratic function which we construct as follows. The dimension of feature vector $\boldsymbol{x}$ is $d$. We set $\mathbf{A} = \mathbf{U}\boldsymbol{\Sigma}\mathbf{U}^\top$, where $\mathbf{U}$ is obtained from QR decomposition of a random matrix with entries sampled independently from $\mathcal{N}(0,1)$, and $\boldsymbol{\Sigma}$ is set as in Table 1. The vector $\boldsymbol{b}$ is generated with entries independently drawn from $\mathcal{N}(0,1)$. For each problem instance, the initial point is randomly initialized from $\mathcal{N}(0,1)$ as well.

The decreasing step sizes for both algorithms are set appropriately. According to the theoretical results of ZSG and ZSC, the step sizes for them are set proportional to $\mathcal{O}(1/(\mathrm{tr}(\mathbf{M}) + \lambda_{\min}(\mathbf{M})t))$ and $\mathcal{O}(1/(d\lambda_{\max}(\mathbf{M}) + \lambda_{\min}(\mathbf{M})t))$, respectively. We report the experimental results in Figure 1.

When $d = 100$, we observe that in the first two experiments, ZSG outperforms ZSC. As $\mathrm{tr}(\mathbf{M})$ increases, the performance of ZSG deteriorates. Nevertheless, ZSG remains superior to ZSC under Skewed Hessian Spectra.

When $d = 500$, we find that ZSG significantly outperforms ZSC in the remaining two experiments. Similar performance trends are observed as $\mathrm{tr}(\mathbf{M})$ increases. Notably, as the problem dimension grows, the eigenvalues of the Hessian matrix become more diverse, making ZSG increasingly advantageous over ZSC. All results are consistent with our theoretical analysis.

### 5.2 LOGISTIC REGRESSION FOR BINARY CLASSIFICATION

In this part, we use real datasets to compare the convergence behavior of ZSG and ZSC on strongly convex problems. We consider logistic regression the following loss function

$$f(\boldsymbol{x}) = \frac{1}{n}\sum_{i=1}^{n}\log[1 + \exp(-y_i\langle \boldsymbol{a}_i, \boldsymbol{x}\rangle)] + \frac{\beta}{2}\|\boldsymbol{x}\|^2,$$

Table 1: Setting of diagonal matrix $\mathbf{\Sigma}$ used to construct $\mathbf{A}$.

| Type | $\mathbf{\Sigma}$ |
|---|---|
| 1 | $d = 100$ Matrix with first 99 components equal to 10 and the remaining one equal to $10\sqrt{10}$ |
| 2 | $d = 100$ Matrix with first 80 components equal to 10 and the rest equal to $10\sqrt{10}$ |
| 3 | $d = 500$ Matrix with first 499 components equal to $10\sqrt{5}$ and the remaining one equal to $100\sqrt{5}$ |
| 4 | $d = 500$ Matrix with first 480 components equal to $10\sqrt{5}$ and the rest equal to $100\sqrt{5}$ |

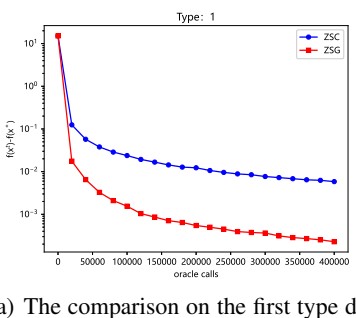
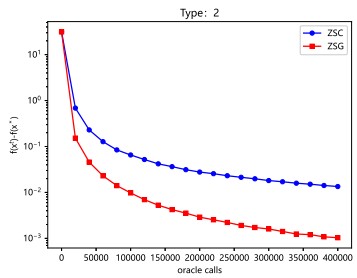

(a) The comparison on the first type diagonal matrix of Table 1

(b) The comparison on the second type diagonal matrix of Table 1

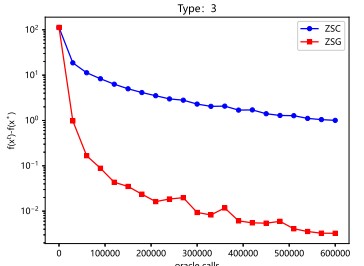
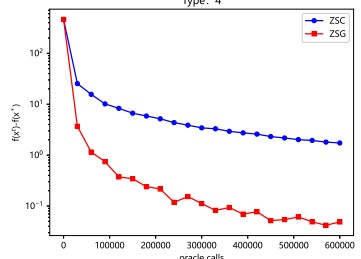

(c) The comparison on the third type diagonal matrix of Table 1

(d) The comparison on the fourth type diagonal matrix of Table 1

Figure 1: Comparison of running results of ZSG and ZSC on quadratic functions.

where $\boldsymbol{a}_i \in \mathbb{R}^d$ denotes the i-th input vector, $y_i \in \{-1, 1\}$ is the corresponding label and $\beta$ is the regularization parameter. We conduct experiments on the 'mushrooms', 'phishing' and 'a8a' datasets, with $d = 112, 68$ and $123$, respectively. All three datasets can be downloaded from libsvm datasets. Through the analysis in Section F, we find that all of these datasets fall into the skewed-Hessian setting. In our experiments on 'mushrooms' and 'phishing', we divide the training set and test set in a ratio of 4:1 and set $\beta = 0.001$. We properly choose the decreasing step sizes of them. We report the experimental results in Figure 2.

The first row of subfigures presents the training loss across all experiments. It can be observed that ZSG outperforms ZSC. The second row displays the corresponding test accuracy. The test accuracy achieved by ZSG is more competitive across all experiments. Therefore, the results consistently demonstrate the superiority of ZSG. Intuitively, ZSG benefits from simultaneously incorporating all coordinates in each oracle call, whereas ZSC estimates gradients along individual coordinates. In addition, we conduct a sensitivity analysis of $\alpha$ in Section G. We find that an large $\alpha$ significantly hinders the convergence process, while $\alpha = 10^{-2}$ is already sufficient for most cases.

## 6 CONCLUSION AND FUTURE WORK

In this paper, we are the first to theoretically analyze the potential advantages of ZSG compared to ZSC and obtain the best result from quadratic functions to a broader class of objective func-

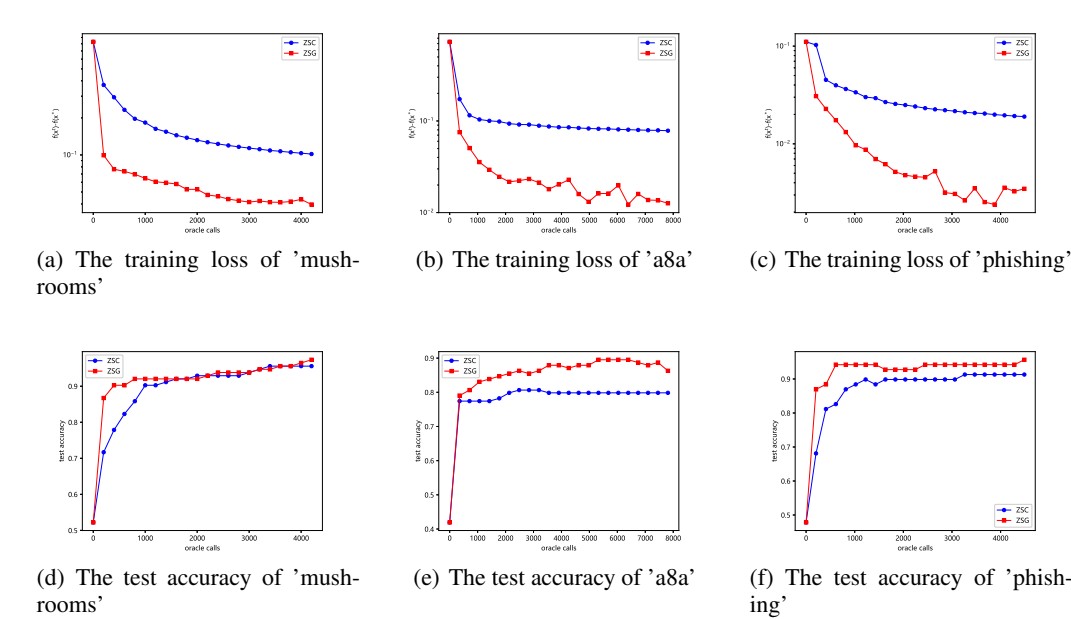

(a) The training loss of 'mush-rooms'

(b) The training loss of 'a8a'

(c) The training loss of 'phishing'

(d) The test accuracy of 'mush-rooms'

(e) The test accuracy of 'a8a'

(f) The test accuracy of 'phish-ing'

Figure 2: Comparison of running results of ZSG and ZSC on binary classification problem.

tions. When $\sigma^2 = 0$ and the objective function is quadratic, we recover the main conclusion proposed by Wang et al. (2024): the complexity of ZSG, given by $\mathcal{O}\left(\frac{\mathrm{tr}(\mathbf{M})}{\lambda_{\min}(\mathbf{M})} \log \frac{1}{\varepsilon}\right)$, outperforms the coordinate-sketched SEGA algorithm, whose complexity is $\mathcal{O}\left(\frac{d\lambda_{\max}(\mathbf{M})}{\lambda_{\min}(\mathbf{M})} \log \frac{1}{\varepsilon}\right)$ in the field of zeroth-order optimization. Notably, leveraging the quadratic regularity assumption, we extend the result of Wang et al. (2024) beyond quadratic functions to a broader class of objectives, and rigorously establish an $\mathcal{O}\left(\frac{\gamma_u \mathrm{tr}(\mathbf{M})}{\gamma_l \lambda_{\min}(\mathbf{M})} \log \frac{1}{\varepsilon}\right)$ complexity guarantee for ZSG. When $\sigma^2 > 0$ and the objective function is quadratic, we establish the main conclusion of our paper: the query complexity of ZSG is $\mathcal{O}\left(\frac{\mathrm{tr}(\mathbf{M})\sigma^2}{\lambda_{\min}^2(\mathbf{M})} \frac{1}{\varepsilon}\right)$, which outperforms ZSC algorithm, whose query complexity is $\mathcal{O}\left(\frac{d\lambda_{\max}(\mathbf{M})\sigma^2}{\lambda_{\min}^2(\mathbf{M})} \frac{1}{\varepsilon}\right)$. We further establish that, even for non-quadratic objectives, ZSG achieves a complexity of $\mathcal{O}\left(\frac{\gamma_u \mathrm{tr}(\mathbf{M})\sigma^2}{\gamma_l^2 \lambda_{\min}^2(\mathbf{M})} \frac{1}{\varepsilon}\right)$, outperforming the $\mathcal{O}\left(\frac{\gamma_u d\lambda_{\max}(\mathbf{M})\sigma^2}{\gamma_l^2 \lambda_{\min}^2(\mathbf{M})} \frac{1}{\varepsilon}\right)$ complexity of ZSC. ZSG also exhibits weak dimensional dependence and demonstrates a notable advantage in practice, primarily skewed Hessian spectra are commonly observed in real-world problems (Yue et al., 2023).

The upper and lower quadratic regularity constants enable us to generalize the results from quadratic functions to a broader class of objective functions, although $\gamma_u$ and $\gamma_l$ are indeed difficult to control. Additional assumptions or a more refined analysis may be required to verify whether $\frac{\gamma_u}{\gamma_l} = \mathcal{O}(1)$ or $\frac{\gamma_u}{\gamma_l^2} = \mathcal{O}(1)$ holds, thereby guiding better parameter tuning strategies in future settings. Frangella et al. (2023) point out that for any objective with a Lipschitz Hessian, $\frac{\gamma_u}{\gamma_l}$ approaches 1 as the optimal objective value is approached. This insight helps explain the empirical advantage of ZSG in the fine-tuning of large language models. Moreover, our experimental evaluation includes not only quadratic objectives but also logistic regression tasks. These results highlight the practical significance of extending our convergence analysis beyond the quadratic setting.

In addition, a promising research direction is to incorporate Hessian information into the gradient estimator $\hat{\nabla} f(\boldsymbol{x}^t, \mathcal{S}_t)$. The motivation stems from the observation that a significant difference in the curvature of the loss function can lead to training instability or slow convergence. Hessian information can be leveraged to adaptively scale parameter updates, thereby mitigating this issue. We expect that integrating such techniques into our analytical framework could lead to improved practical performance in terms of query complexity.

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

## A   THE USE OF LARGE LANGUAGE MODELS (LLMs)

This paper uses large language models only to polish the language and adjust the paragraph structure.

## B   SEVERAL USEFUL LEMMAS

In this section, we introduce several useful lemmas. The following lemma shows that the expectation of the product of two quadratic forms of the random Gaussian vector is related to the trace of the corresponding matrix.

**Lemma B.1** (Magnus et al. (1978)). *Let $\mathbf{A}$ and $\mathbf{B}$ be two symmetric matrices, and $\boldsymbol{u}$ obeys the Gaussian distribution, that is, $\boldsymbol{u} \sim \mathcal{N}(\mathbf{0}, \mathbf{I}_d)$. Define $z = \boldsymbol{u}^\top \mathbf{A} \boldsymbol{u} \cdot \boldsymbol{u}^\top \mathbf{B} \boldsymbol{u}$. The expectation of $z$ is*

$$\mathbb{E}_{\boldsymbol{u}}[z] = (\mathrm{tr}\mathbf{A})(\mathrm{tr}\mathbf{B}) + 2(\mathrm{tr}\mathbf{A}\mathbf{B}). \tag{15}$$

**Lemma B.2** (Nesterov & Spokoiny (2017)). *Let $\boldsymbol{u}$ obeys the Gaussian distribution, that is, $\boldsymbol{u} \sim \mathcal{N}(\mathbf{0}, \mathbf{I}_d)$. We define normalization constant $\kappa = \int e^{-\frac{1}{2}\|\boldsymbol{u}\|^2} d\boldsymbol{u}$ and define moments $\mathbf{M}_p = \frac{1}{\kappa} \int \|\boldsymbol{u}\|^p e^{-\frac{1}{2}\|\boldsymbol{u}\|^2} d\boldsymbol{u}$. For $p \geq 2$, we can obtain upper bounds*

$$n^{p/2} \leq \mathbf{M}_p \leq (p+d)^{p/2}. \tag{16}$$

**Lemma B.3.** *If we have a positive definite matrix $\mathbf{M}$ defined as weighted inner product, for all $\boldsymbol{x} \in \mathbb{R}^d$, we can obtain the following inequalities*

$$\|\boldsymbol{x}\|_{\mathbf{M}}^2 \leq \mathrm{tr}(\mathbf{M}) \|\boldsymbol{x}\|^2, \tag{17}$$

$$\lambda_{\min}(\mathbf{M}) \|\boldsymbol{x}\|^2 \leq \|\boldsymbol{x}\|_{\mathbf{M}}^2 \leq \lambda_{\max}(\mathbf{M}) \|\boldsymbol{x}\|^2. \tag{18}$$

*Proof.* For a positive definite matrix $\mathbf{M}$, there must exist an orthogonal matrix $\mathbf{T}$ such that $\mathbf{M}$ is similar to a diagonal matrix whose elements are eigenvalues of matrix $\mathbf{M}$. We denote $\lambda_i$ be the i-th eigenvalue of matrix $\mathbf{M}$, then, we can obtain an equation as follows

$$\mathbf{M} = \mathbf{T}\mathrm{diag}\left\{\lambda_1, \lambda_2, \cdots, \lambda_d\right\} \mathbf{T}^{-1}. \tag{19}$$

Let $\boldsymbol{y} = \mathbf{T}^\top \boldsymbol{x}$, then, we can easily prove this Lemma. We first prove Eq. (17)

$$
\begin{aligned}
\|\boldsymbol{x}\|_{\mathbf{M}}^2 =& \langle \mathbf{M}\boldsymbol{x}, \boldsymbol{x} \rangle = \boldsymbol{x}^\top \mathbf{M}\boldsymbol{x} \overset{(19)}{=} x^\top \mathbf{T}\mathrm{diag}\left\{\lambda_1, \lambda_2, \cdots, \lambda_d\right\} \mathbf{T}^{-1}x \\
=& \boldsymbol{x}^\top \mathbf{T}\mathrm{diag}\left\{\lambda_1, \lambda_2, \cdots, \lambda_d\right\} \mathbf{T}^\top \boldsymbol{x} \\
=& \boldsymbol{y}^\top \mathrm{diag}\left\{\lambda_1, \lambda_2, \cdots, \lambda_d\right\} \boldsymbol{y} \\
\leq& \mathrm{tr}(\mathbf{M})\boldsymbol{x}^\top \mathbf{T}\mathbf{T}^\top \boldsymbol{x} \\
=& \mathrm{tr}(\mathbf{M}) \|\boldsymbol{x}\|^2.
\end{aligned}
$$

Similarly, we can prove the Eq. (18). $\qquad\square$

**Lemma B.4.** *For the sake of simplicity in the subsequent proof, we first derive the upper bound of $\hat{\nabla} f(\boldsymbol{x}^t, \mathcal{S}_t)$. The upper bound is related to $\nabla f(\boldsymbol{x}^t, \mathcal{S}_t)$ and $\alpha$:*

$$\mathbb{E}_{\boldsymbol{u}_t} \left[ \|\hat{\nabla} f(\boldsymbol{x}^t, \mathcal{S}_t)\|_{\mathbf{M}(\boldsymbol{z}^t)}^2 \right] \leq 6\mathrm{tr}(\mathbf{M}(\boldsymbol{z}^t)) \|\nabla f(\boldsymbol{x}^t, \mathcal{S}_t)\|^2 + \frac{(6+d)^3 \gamma_u^2 \alpha^2}{2}. \tag{20}$$

*Proof.* This part of the proof involves the basic properties of the norm and some important lemmas.

$$
\begin{aligned}
\mathbb{E}_{\boldsymbol{u}_t}\left[\|\hat{\nabla}f(\boldsymbol{x}^t,\mathcal{S}_t)\|_{\mathbf{M}(\boldsymbol{z}^t)}^2\right] &\overset{(5)}{=} \mathbb{E}_{\boldsymbol{u}_t}\left[\left\|\boldsymbol{u}_t\boldsymbol{u}_t^\top\nabla f(\boldsymbol{x}^t,\mathcal{S}_t)+\phi(\boldsymbol{u}_t,\alpha,\boldsymbol{x}^t)\right\|_{\mathbf{M}(\boldsymbol{z}^t)}^2\right] \\
&\leq 2\mathbb{E}_{\boldsymbol{u}_t}\left[\left\|\boldsymbol{u}_t\boldsymbol{u}_t^\top\nabla f(\boldsymbol{x}^t,\mathcal{S}_t)\right\|_{\mathbf{M}(\boldsymbol{z}^t)}^2\right]+2\mathbb{E}_{\boldsymbol{u}_t}\left[\left\|\phi(\boldsymbol{u}_t,\alpha,\boldsymbol{x}^t)\right\|_{\mathbf{M}(\boldsymbol{z}^t)}^2\right] \\
&\overset{(6)}{\leq} 2\mathbb{E}_{\boldsymbol{u}_t}\left[\left\|\boldsymbol{u}_t\boldsymbol{u}_t^\top\nabla f(\boldsymbol{x}^t,\mathcal{S}_t)\right\|_{\mathbf{M}(\boldsymbol{z}^t)}^2\right]+\frac{\gamma_u^2\alpha^2}{2}\mathbb{E}_{\boldsymbol{u}_t}\left[\|\boldsymbol{u}_t\|_{\mathbf{M}(\boldsymbol{z}^t)}^6\right] \\
&\overset{(8)+(16)+(18)}{\leq} 6\mathrm{tr}(\mathbf{M}(\boldsymbol{z}^t))\left\|\nabla f(\boldsymbol{x}^t,\mathcal{S}_t)\right\|^2+\frac{\lambda_{\max}^3(\mathbf{M}(\boldsymbol{z}^t))(6+d)^3\gamma_u^2\alpha^2}{2}.
\end{aligned}
$$

$\square$

**Lemma B.5.** *For the sake of simplicity in the subsequent proof, we will derive the upper bound of an important inner product $\langle\nabla f(\boldsymbol{x}^t),\phi(\boldsymbol{u}_t,\alpha)\rangle$. The upper bound is related to real gradient $\nabla f(\boldsymbol{x}^t)$ and $\alpha$:*

$$
-\mathbb{E}_{\boldsymbol{u}_t}\left[\langle\nabla f(\boldsymbol{x}^t),\phi(\boldsymbol{u}_t,\alpha,\boldsymbol{x}^t)\rangle\right] \leq \frac{1}{2}\left\|\nabla f(\boldsymbol{x}^t)\right\|^2+\frac{\lambda_{\max}^2(\mathbf{M}(\boldsymbol{z}^t))(6+d)^3\gamma_u^2\alpha^2}{8}. \tag{21}
$$

*Proof.* The techniques involved in this part are similar to those in Lemma B.4.

$$
\begin{aligned}
-\mathbb{E}_{\boldsymbol{u}_t}\left[\langle\nabla f(\boldsymbol{x}^t),\phi(\boldsymbol{u}_t,\alpha,\boldsymbol{x}^t)\rangle\right] &\leq \mathbb{E}_{\boldsymbol{u}_t}\left[\left\|\nabla f(\boldsymbol{x}^t)\right\|\left\|\phi(\boldsymbol{u}_t,\alpha,\boldsymbol{x}^t)\right\|\right] \\
&\leq \frac{1}{2}\left\|\nabla f(\boldsymbol{x}^t)\right\|^2+\frac{1}{2}\mathbb{E}_{\boldsymbol{u}_t}\left[\left\|\phi(\boldsymbol{u}_t,\alpha,\boldsymbol{x}^t)\right\|^2\right] \\
&\overset{(6)}{\leq} \frac{1}{2}\left\|\nabla f(\boldsymbol{x}^t)\right\|^2+\frac{\gamma_u^2\alpha^2}{8}\mathbb{E}_{\boldsymbol{u}_t}\left[\|\boldsymbol{u}_t\|_{\mathbf{M}(\boldsymbol{z}^t)}^4\cdot\|\boldsymbol{u}_t\|^2\right] \\
&\overset{(18)}{\leq} \frac{1}{2}\left\|\nabla f(\boldsymbol{x}^t)\right\|^2+\frac{\lambda_{\max}^2(\mathbf{M}(\boldsymbol{z}^t))\gamma_u^2\alpha^2}{8}\mathbb{E}_{\boldsymbol{u}_t}\left[\|\boldsymbol{u}_t\|^6\right] \\
&\overset{(16)}{\leq} \frac{1}{2}\left\|\nabla f(\boldsymbol{x}^t)\right\|^2+\frac{\lambda_{\max}^2(\mathbf{M}(\boldsymbol{z}^t))(6+d)^3\gamma_u^2\alpha^2}{8}.
\end{aligned}
$$

$\square$

# C  PROOF OF IMPORTANT LEMMAS

In this section, we give some details of proof about some important Lemmas.

## C.1  PROOF OF LEMMA 3.1

*Proof.* By the Taylor's expansion, we can obtain that

$$
f(\boldsymbol{x}+\alpha\boldsymbol{u},\mathcal{S})=f(\boldsymbol{x})+\alpha\langle\nabla f(\boldsymbol{x},\mathcal{S}),\boldsymbol{u}\rangle+\phi'(\boldsymbol{u},\alpha,\boldsymbol{x})
$$

where $\phi'(\boldsymbol{u},\alpha,\boldsymbol{x})=f(\boldsymbol{x}+\alpha\boldsymbol{u},\mathcal{S})-f(\boldsymbol{x})-\alpha\langle\nabla f(\boldsymbol{x},\mathcal{S}),\boldsymbol{u}\rangle$. Similarly, we can obtain

$$
f(\boldsymbol{x}-\alpha\boldsymbol{u},\mathcal{S})=f(\boldsymbol{x})-\alpha\langle\nabla f(\boldsymbol{x},\mathcal{S}),\boldsymbol{u}\rangle+\phi'(\boldsymbol{u},-\alpha,\boldsymbol{x}).
$$

$$
\hat{\nabla}f(\boldsymbol{x},\mathcal{S})=\frac{[f(\boldsymbol{x}+\alpha\boldsymbol{u},\mathcal{S})-f(\boldsymbol{x}-\alpha\boldsymbol{u},\mathcal{S})]}{2\alpha}\cdot\boldsymbol{u}=\boldsymbol{u}\boldsymbol{u}^\top\nabla f(\boldsymbol{x},\mathcal{S})+\frac{\phi'(\boldsymbol{u},\alpha,\boldsymbol{x})-\phi'(\boldsymbol{u},-\alpha,\boldsymbol{x})}{2\alpha}\cdot\boldsymbol{u}.
$$

By the upper quadratically regular assumption, we can obtain that

$$
|\phi'(\boldsymbol{u},\alpha,\boldsymbol{x})|=|f(\boldsymbol{x}+\alpha\boldsymbol{u},\mathcal{S})-f(\boldsymbol{x})-\alpha\langle\nabla f(\boldsymbol{x},\mathcal{S}),\boldsymbol{u}\rangle|\leq\frac{\gamma_u\alpha^2}{2}\|\boldsymbol{u}\|_{\mathbf{M}(\boldsymbol{z}_1)}^2,
$$

$$
|\phi'(\boldsymbol{u},-\alpha,\boldsymbol{x})|=|f(\boldsymbol{x}-\alpha\boldsymbol{u},\mathcal{S})-f(\boldsymbol{x})+\alpha\langle\nabla f(\boldsymbol{x},\mathcal{S}),\boldsymbol{u}\rangle|\leq\frac{\gamma_u\alpha^2}{2}\|\boldsymbol{u}\|_{\mathbf{M}(\boldsymbol{z}_2)}^2.
$$

Then, we can finally obtain that

$$
\left\|\frac{\phi'(\boldsymbol{u},\alpha,\boldsymbol{x})-\phi'(\boldsymbol{u},-\alpha,\boldsymbol{x})}{2\alpha}\cdot\boldsymbol{u}\right\|\leq\frac{|\phi'(\boldsymbol{u},\alpha,\boldsymbol{x})|+|\phi'(\boldsymbol{u},-\alpha,\boldsymbol{x})|}{2\alpha}\|\boldsymbol{u}\|\leq\frac{\gamma_u\alpha}{2}\|\boldsymbol{u}\|_{\mathbf{M}(\boldsymbol{z})}^2\cdot\|\boldsymbol{u}\|.
$$

$\square$

## C.2 PROOF OF LEMMA 4.1

*Proof.* This part of the proof mainly relies on the properties of the matrix trace.

$$
\begin{aligned}
\mathbb{E}_{\boldsymbol{u}_t}\left[\left\|\boldsymbol{u}_t\boldsymbol{u}_t^\top\nabla f(\boldsymbol{x}^t,\mathcal{S}_t)\right\|_{\mathbf{M}}^2\right] =&\mathbb{E}_{\boldsymbol{u}_t}\left[\nabla f(\boldsymbol{x}^t,\mathcal{S}_t)^\top\boldsymbol{u}_t\boldsymbol{u}_t^\top\mathbf{M}^\top\boldsymbol{u}_t\boldsymbol{u}_t^\top\nabla f(\boldsymbol{x}^t,\mathcal{S}_t)\right]\\
=&\mathbb{E}_{\boldsymbol{u}_t}\left[\operatorname{tr}(\nabla f(\boldsymbol{x}^t,\mathcal{S}_t)^\top\boldsymbol{u}_t\boldsymbol{u}_t^\top\mathbf{M}^\top\boldsymbol{u}_t\boldsymbol{u}_t^\top\nabla f(\boldsymbol{x}^t,\mathcal{S}_t))\right]\\
=&\mathbb{E}_{\boldsymbol{u}_t}\left[\operatorname{tr}(\boldsymbol{u}_t^\top\mathbf{M}^\top\boldsymbol{u}_t\boldsymbol{u}_t^\top\nabla f(\boldsymbol{x}^t,\mathcal{S}_t)\nabla f(\boldsymbol{x}^t,\mathcal{S}_t)^\top\boldsymbol{u}_t)\right]\\
\overset{(15)}{=}&\operatorname{tr}(\mathbf{M})\operatorname{tr}(\nabla f(\boldsymbol{x}^t,\mathcal{S}_t)\nabla f(\boldsymbol{x}^t,\mathcal{S}_t)^\top)\\
&+2\operatorname{tr}(\nabla f(\boldsymbol{x}^t,\mathcal{S}_t)^\top\mathbf{M}^\top\nabla f(\boldsymbol{x}^t,\mathcal{S}_t))\\
=&\operatorname{tr}(\mathbf{M})\left\|\nabla f(\boldsymbol{x}^t,\mathcal{S}_t)\right\|^2+2\left\|\nabla f(\boldsymbol{x}^t,\mathcal{S}_t)\right\|_{\mathbf{M}}^2\\
\overset{(17)}{\leq}&3\operatorname{tr}(\mathbf{M})\left\|\nabla f(\boldsymbol{x}^t,\mathcal{S}_t)\right\|^2.
\end{aligned}
$$

$\square$

## C.3 PROOF OF LEMMA 4.2

*Proof.* We use the lower quadratically regular introduced in Assumption 2.1,

$$
f(\boldsymbol{y}) \geq f(\boldsymbol{x}) + \langle\nabla f(\boldsymbol{x}), \boldsymbol{y}-\boldsymbol{x}\rangle + \frac{\gamma_l}{2}\|\boldsymbol{y}-\boldsymbol{x}\|_{\mathbf{M}(\boldsymbol{z})}^2.
$$

Then, we construct an auxiliary function,

$$
F(\boldsymbol{y}) = f(\boldsymbol{x}) + \langle\nabla f(\boldsymbol{x}), \boldsymbol{y}-\boldsymbol{x}\rangle + \frac{\gamma_l}{2}\|\boldsymbol{y}-\boldsymbol{x}\|_{\mathbf{M}(\boldsymbol{z})}^2.
$$

To obtain the minimum of the auxiliary function, we need to make

$$
\nabla F(\boldsymbol{y}^*) = \nabla f(\boldsymbol{x}) + 2\gamma_l\mathbf{M}(\boldsymbol{z})(\boldsymbol{y}^*-\boldsymbol{x}) = 0.
$$

So, we can find that

$$
\boldsymbol{y}^* = \boldsymbol{x} - \frac{1}{\gamma_l}\mathbf{M}(\boldsymbol{z})^{-1}\nabla f(\boldsymbol{x}). \tag{22}
$$

Using the above information, we can continue to deduce that

$$
\begin{aligned}
f(\boldsymbol{y}) \geq& F(\boldsymbol{y})\\
\geq& F(\boldsymbol{y}^*)\\
\overset{(22)}{=}& f(\boldsymbol{x}) - \left\langle\nabla f(\boldsymbol{x}), \frac{1}{\gamma_l}\mathbf{M}(\boldsymbol{z})^{-1}\nabla f(\boldsymbol{x})\right\rangle + \frac{\gamma_l}{2}\left\|\frac{1}{\gamma_l}\mathbf{M}(\boldsymbol{z})^{-1}\nabla f(\boldsymbol{x})\right\|_{\mathbf{M}(\boldsymbol{z})}^2\\
=& f(\boldsymbol{x}) - \frac{1}{\gamma_l}\|\nabla f(\boldsymbol{x})\|_{\mathbf{M}(\boldsymbol{z})^{-1}}^2 + \frac{1}{2\gamma_l}\nabla f(\boldsymbol{x})^\top(\mathbf{M}(\boldsymbol{z})^{-1})^\top\mathbf{M}(\boldsymbol{z})^\top\mathbf{M}(\boldsymbol{z})^{-1}\nabla f(\boldsymbol{x})\\
=& f(\boldsymbol{x}) - \frac{1}{\gamma_l}\|\nabla f(\boldsymbol{x})\|_{\mathbf{M}(\boldsymbol{z})^{-1}}^2 + \frac{1}{2\gamma_l}\|\nabla f(\boldsymbol{x})\|_{\mathbf{M}(\boldsymbol{z})^{-1}}^2\\
=& f(\boldsymbol{x}) - \frac{1}{2\gamma_l}\|\nabla f(\boldsymbol{x})\|_{\mathbf{M}(\boldsymbol{z})^{-1}}^2.
\end{aligned}
$$

Let $\boldsymbol{x} = \boldsymbol{x}^t, \boldsymbol{y} = \boldsymbol{x}^*$, and rearrange the above formula, we can obtain

$$
\begin{aligned}
f(\boldsymbol{x}^t) - f^* \leq& \frac{1}{2\gamma_l}\left\|\nabla f(\boldsymbol{x}^t)\right\|_{\mathbf{M}(\boldsymbol{z})^{-1}}^2\\
\overset{(18)}{\leq}& \frac{\lambda_{\max}(\mathbf{M}(\boldsymbol{z})^{-1})}{2\gamma_l}\left\|\nabla f(\boldsymbol{x}^t)\right\|\\
=& \frac{1}{2\gamma_l\lambda_{\min}(\mathbf{M}(\boldsymbol{z}))}\left\|\nabla f(\boldsymbol{x}^t)\right\|.
\end{aligned}
$$

$\square$

# D PROOF OF MAIN THEOREMS

In this section, we give some details of proof about some important Theorem.

## D.1 PROOF OF THEOREM 4.3

*Proof.* Firstly, we can deduce the expectation of $f(\boldsymbol{x}^{t+1})$,

$$
\begin{aligned}
f(\boldsymbol{x}^{t+1}) &\overset{(2)}{\leq} f(\boldsymbol{x}^t) + \left\langle \nabla f(\boldsymbol{x}^t), \boldsymbol{x}^{t+1} - \boldsymbol{x}^t \right\rangle + \frac{1}{2} \left\| \boldsymbol{x}^{t+1} - \boldsymbol{x}^t \right\|_{\mathbf{M}}^2 \\
&\overset{(7)+(5)}{=} f(\boldsymbol{x}^t) - \eta \left\langle \nabla f(\boldsymbol{x}^t), \boldsymbol{u}_t \boldsymbol{u}_t^\top \nabla f(\boldsymbol{x}, \mathcal{S}) + \phi(\boldsymbol{u}_t, \alpha, \boldsymbol{x}^t) \right\rangle \\
&\quad + \frac{\eta^2}{2} \left\| \boldsymbol{u}_t \boldsymbol{u}_t^\top \nabla f(\boldsymbol{x}, \mathcal{S}) + \phi(\boldsymbol{u}_t, \alpha, \boldsymbol{x}^t) \right\|_{\mathbf{M}}^2 .
\end{aligned}
\tag{23}
$$

Let us deduce the expectation of $f(\boldsymbol{x}^{t+1})$ for $u$,

$$
\begin{aligned}
\mathbb{E}_{\boldsymbol{u}_t}\left[ f(\boldsymbol{x}^{t+1}) \right] =& f(\boldsymbol{x}^t) - \eta \left\langle \nabla f(\boldsymbol{x}^t), \mathbb{E}_{\boldsymbol{u}_t}\left[ \boldsymbol{u}_t \boldsymbol{u}_t^\top \nabla f(\boldsymbol{x}, \mathcal{S}) + \phi(\boldsymbol{u}_t, \alpha, \boldsymbol{x}^t) \right] \right\rangle \\
&+ \frac{\eta^2}{2} \mathbb{E}_{\boldsymbol{u}_t}\left[ \left\| \boldsymbol{u}_t \boldsymbol{u}_t^\top \nabla f(\boldsymbol{x}, \mathcal{S}) + \phi(\boldsymbol{u}_t, \alpha, \boldsymbol{x}^t) \right\|_{\mathbf{M}}^2 \right] \\
\leq& f(\boldsymbol{x}^t) - \eta \left\langle \nabla f(\boldsymbol{x}^t), \nabla f(\boldsymbol{x}^t, \mathcal{S}_t) \right\rangle - \eta \mathbb{E}_{\boldsymbol{u}_t}\left[ \left\langle \nabla f(\boldsymbol{x}^t), \phi(\boldsymbol{u}_t, \alpha, \boldsymbol{x}^t) \right\rangle \right] \\
&+ \frac{\eta^2}{2} \mathbb{E}_{\boldsymbol{u}_t}\left[ \left\| \boldsymbol{u}_t \boldsymbol{u}_t^\top \nabla f(\boldsymbol{x}, \mathcal{S}) + \phi(\boldsymbol{u}_t, \alpha, \boldsymbol{x}^t) \right\|_{\mathbf{M}}^2 \right] \\
\overset{(20)+(21)}{\leq}& f(\boldsymbol{x}^t) - \eta \left\langle \nabla f(\boldsymbol{x}^t), \nabla f(\boldsymbol{x}^t, \mathcal{S}_t) \right\rangle + \frac{\eta}{2} \left\| \nabla f(\boldsymbol{x}^t) \right\|^2 + 3\eta^2 \mathrm{tr}(\mathbf{M}) \left\| \nabla f(\boldsymbol{x}^t, \mathcal{S}_t) \right\|^2 \\
&+ \frac{\left[ 1 + 2\lambda_{\max}(\mathbf{M})\eta \right] \lambda_{\max}^2(\mathbf{M})(6+d)^3 \alpha^2 \eta}{8} .
\end{aligned}
$$

Then, let us deduce the expectation of $\mathbb{E}_u\left[ f(\boldsymbol{x}^{t+1}) \right]$,

$$
\begin{aligned}
\mathbb{E}\left[ f(\boldsymbol{x}^{t+1}) \right] \leq& f(\boldsymbol{x}^t) - \eta \left\langle \nabla f(\boldsymbol{x}^t), \mathbb{E}\left[ \nabla f(\boldsymbol{x}^t, \mathcal{S}_t) \right] \right\rangle + \frac{\eta}{2} \left\| \nabla f(\boldsymbol{x}^t) \right\|^2 \\
&+ 3\eta^2 \mathrm{tr}(\mathbf{M}) \mathbb{E}\left[ \left\| \nabla f(\boldsymbol{x}^t, \mathcal{S}_t) \right\|^2 \right] + \frac{\left[ 1 + 2\lambda_{\max}(\mathbf{M})\eta \right] \lambda_{\max}^2(\mathbf{M})(6+d)^3 \alpha^2 \eta}{8} \\
=& f(\boldsymbol{x}^t) - \frac{\eta}{2} \left\| \nabla f(\boldsymbol{x}^t) \right\|^2 + 3\eta^2 \mathrm{tr}(\mathbf{M}) \mathbb{E}\left[ \left\| \nabla f(\boldsymbol{x}^t, \mathcal{S}_t) \right\|^2 \right] \\
&+ \frac{\left[ 1 + 2\lambda_{\max}(\mathbf{M})\eta \right] \lambda_{\max}^2(\mathbf{M})(6+d)^3 \alpha^2 \eta}{8} \\
\overset{(4)}{\leq}& f(\boldsymbol{x}^t) - \frac{\eta}{2} \left\| \nabla f(\boldsymbol{x}^t) \right\|^2 + 3\eta^2 \mathrm{tr}(\mathbf{M})(\sigma^2 + \left\| \nabla f(\boldsymbol{x}^t) \right\|^2) \\
&+ \frac{\left[ 1 + 2\lambda_{\max}(\mathbf{M})\eta \right] \lambda_{\max}^2(\mathbf{M})(6+d)^3 \alpha^2 \eta}{8} \\
=& f(\boldsymbol{x}^t) + \left[ 3\eta^2 \mathrm{tr}(\mathbf{M}) - \frac{\eta}{2} \right] \left\| \nabla f(\boldsymbol{x}^t) \right\|^2 + 3\eta^2 \sigma^2 \mathrm{tr}(\mathbf{M}) \\
&+ \frac{\left[ 1 + 2\lambda_{\max}(\mathbf{M})\eta \right] \lambda_{\max}^2(\mathbf{M})(6+d)^3 \alpha^2 \eta}{8} \\
=& f(\boldsymbol{x}^t) + 3\eta^2 \sigma^2 \mathrm{tr}(\mathbf{M}) + \frac{\left[ 1 + 2\lambda_{\max}(\mathbf{M})\eta \right] \lambda_{\max}^2(\mathbf{M})(6+d)^3 \alpha^2 \eta}{8} \\
&- \frac{\eta}{2} \left[ 1 - 6\eta \mathrm{tr}(\mathbf{M}) \right] \left\| \nabla f(\boldsymbol{x}^t) \right\|^2 \\
\overset{(9)}{\leq}& f(\boldsymbol{x}^t) + 3\eta^2 \sigma^2 \mathrm{tr}(\mathbf{M}) + \frac{\left[ 1 + 2\lambda_{\max}(\mathbf{M})\eta \right] \lambda_{\max}^2(\mathbf{M})(6+d)^3 \alpha^2 \eta}{8} \\
&- \eta \lambda_{\min}(\mathbf{M}) \left[ 1 - 6\eta \mathrm{tr}(\mathbf{M}) \right] (f(\boldsymbol{x}^t) - f^*) \\
\leq& f(\boldsymbol{x}^t) + 3\eta^2 \sigma^2 \mathrm{tr}(\mathbf{M}) + \frac{\left[ 1 + 2\lambda_{\max}(\mathbf{M})\eta \right] \lambda_{\max}^2(\mathbf{M})(6+d)^3 \alpha^2 \eta}{8} \\
&- \frac{1}{2} \eta \lambda_{\min}(\mathbf{M})(f(\boldsymbol{x}^t) - f^*).
\end{aligned}
\tag{24}
$$

And then, let us use the optimal value $f^*$ to transform the inequality,

$$\mathbb{E}\left[f(\boldsymbol{x}^{t+1}) - f^*\right] + f^* - f(\boldsymbol{x}^t) \leq 3\eta^2\sigma^2\mathrm{tr}(\mathbf{M}) + \frac{\left[1 + 2\lambda_{\max}(\mathbf{M})\eta\right]\lambda_{\max}^2(\mathbf{M})(6+d)^3\alpha^2\eta}{8}$$
$$-\frac{1}{2}\eta\lambda_{\min}(\mathbf{M})\mathbb{E}\left[f(\boldsymbol{x}^t) - f^*\right].$$

Rearranging the above formula, we can obtain,

$$\mathbb{E}\left[f(\boldsymbol{x}^{t+1}) - f^*\right] \leq 3\eta^2\sigma^2\mathrm{tr}(\mathbf{M}) + \frac{\left[1 + 2\lambda_{\max}(\mathbf{M})\eta\right]\lambda_{\max}^2(\mathbf{M})(6+d)^3\alpha^2\eta}{8}$$
$$+\left[1 - \frac{1}{2}\eta\lambda_{\min}(\mathbf{M})\right]\mathbb{E}\left[f(\boldsymbol{x}^t) - f^*\right].$$

We need to construct a recursive relation with the following structure,

$$\mathbb{E}\left[f(\boldsymbol{x}^{t+1}) - f^* - \beta\right] \leq \left[1 - \frac{1}{2}\eta\lambda_{\min}(\mathbf{M})\right]\mathbb{E}\left[f(x^t) - f^* - \beta\right].$$

If $\beta = \dfrac{24\eta\mathrm{tr}(\mathbf{M})\sigma^2 + \left[1 + 2\lambda_{\max}(\mathbf{M})\eta\right]\lambda_{\max}^2(\mathbf{M})(6+d)^3\alpha^2}{4\lambda_{\min}(\mathbf{M})}$, the above formula can be derived as

$$\mathbb{E}\left[f(\boldsymbol{x}^{t+1}) - f^* - \frac{24\eta\mathrm{tr}(\mathbf{M})\sigma^2 + \left[1 + 2\lambda_{\max}(\mathbf{M})\eta\right]\lambda_{\max}^2(\mathbf{M})(6+d)^3\alpha^2}{4\lambda_{\min}(\mathbf{M})}\right]$$
$$\leq \left[1 - \frac{1}{2}\eta\lambda_{\min}(\mathbf{M})\right]\mathbb{E}\left[f(x^t) - f^* - \frac{24\eta\mathrm{tr}(\mathbf{M})\sigma^2 + \left[1 + 2\lambda_{\max}(\mathbf{M})\eta\right]\lambda_{\max}^2(\mathbf{M})(6+d)^3\alpha^2}{4\lambda_{\min}(\mathbf{M})}\right]$$
$$\leq \left[1 - \frac{1}{2}\eta\lambda_{\min}(\mathbf{M})\right]^t\left[f(x^0) - f^* - \frac{24\eta\mathrm{tr}(\mathbf{M})\sigma^2 + \left[1 + 2\lambda_{\max}(\mathbf{M})\eta\right]\lambda_{\max}^2(\mathbf{M})(6+d)^3\alpha^2}{4\lambda_{\min}(\mathbf{M})}\right]$$
$$\leq \left[1 - \frac{1}{2}\eta\lambda_{\min}(\mathbf{M})\right]^t\left[f(x^0) - f^*\right].$$

Thus, we can obtain that

$$\mathbb{E}\left[f(\boldsymbol{x}^{t+1}) - f^*\right] \leq \frac{24\eta\mathrm{tr}(\mathbf{M})\sigma^2 + \left[1 + 2\lambda_{\max}(\mathbf{M})\eta\right]\lambda_{\max}^2(\mathbf{M})(6+d)^3\alpha^2}{4\lambda_{\min}(\mathbf{M})}$$
$$+\left[1 - \frac{1}{2}\eta\lambda_{\min}(\mathbf{M})\right]^t\left[f(x^0) - f^*\right].$$

$\square$

### D.2 PROOF OF THEOREM 4.5

*Proof.* Firstly, if we choose decreasing step size $\eta_t$, based on D.1, we can obtain the following formula

$$\mathbb{E}\left[f(\boldsymbol{x}^{t+1}) - f(\boldsymbol{x}^t)\right] \leq 3\eta_t^2\sigma^2\mathrm{tr}(\mathbf{M}) + \frac{\left[1 + 2\lambda_{\max}(\mathbf{M})\eta_t\right]\lambda_{\max}^2(\mathbf{M})(6+d)^3\alpha^2\eta_t}{8}$$
$$-\eta_t\lambda_{\min}(\mathbf{M})\left[1 - 6\eta_t\mathrm{tr}(\mathbf{M})\right]\mathbb{E}\left[f(\boldsymbol{x}^t) - f^*\right]$$
$$\leq 3\eta_t^2\sigma^2\mathrm{tr}(\mathbf{M}) + \frac{\left[1 + 2\lambda_{\max}(\mathbf{M})\eta_t\right]\lambda_{\max}^2(\mathbf{M})(6+d)^3\alpha^2\eta_t}{8}$$
$$-\eta_t\lambda_{\min}(\mathbf{M})\left[1 - 6\eta_0\mathrm{tr}(\mathbf{M})\right]\mathbb{E}\left[f(\boldsymbol{x}^t) - f^*\right]$$
$$\leq 3\eta_t^2\sigma^2\mathrm{tr}(\mathbf{M}) + \frac{\left[1 + 2\lambda_{\max}(\mathbf{M})\eta_t\right]\lambda_{\max}^2(\mathbf{M})(6+d)^3\alpha^2\eta_t}{8}$$
$$-\frac{1}{2}\eta_t\lambda_{\min}(\mathbf{M})\mathbb{E}\left[f(\boldsymbol{x}^t) - f^*\right].$$

Let us prove the final result by induction, for $t = 0$

$$\mathbb{E}\left[f(\boldsymbol{x}^0) - f^*\right] = f(\boldsymbol{x}^0) - f^* = \frac{\gamma}{\gamma + 0}\left[f(\boldsymbol{x}^0) - f^*\right] \leq \frac{v}{\gamma + 0},$$

by the definition of $v$.

Suppose that holds for $t > 0$, then

$$
\begin{aligned}
\mathbb{E}\left[f(\boldsymbol{x}^{t+1}) - f^*\right] \leq & 3\eta_t^2\sigma^2\mathrm{tr}(\mathbf{M}) + \frac{\left[1 + 2\lambda_{\max}(\mathbf{M})\eta_t\right]\lambda_{\max}^2(\mathbf{M})(6+d)^3\alpha^2\eta_t}{8} \\
& + \left[1 - \frac{1}{2}\eta_t\lambda_{\min}(\mathbf{M})\right]\mathbb{E}\left[f(\boldsymbol{x}^t) - f^*\right] \\
\leq & 3\eta_t^2\sigma^2\mathrm{tr}(\mathbf{M}) + \frac{\left[1 + 2\lambda_{\max}(\mathbf{M})\eta_t\right]\lambda_{\max}^2(\mathbf{M})(6+d)^3\alpha^2\eta_t}{8} \\
& + \left[1 - \frac{1}{2}\eta_t\lambda_{\min}(\mathbf{M})\right]\frac{v}{\gamma+t} \\
= & \frac{3\sigma^2 l^2\mathrm{tr}(\mathbf{M})}{(\gamma+t)^2} + \frac{\lambda_{\max}^2(\mathbf{M})(6+d)^3\alpha^2 l}{8(\gamma+t)} + \frac{\lambda_{\max}^3(\mathbf{M})(6+d)^3\alpha^2 l^2}{4(\gamma+t)^2} \\
& + \left[1 - \frac{l\lambda_{\min}(\mathbf{M})}{2(\gamma+t)}\right]\frac{v}{\gamma+t} \\
= & \frac{(\gamma+t-1)v}{(\gamma+t)^2} + \frac{3\sigma^2 l^2\mathrm{tr}(\mathbf{M})}{(\gamma+t)^2} + \frac{\lambda_{\max}^2(\mathbf{M})(6+d)^3\alpha^2 l}{8(\gamma+t)} \\
& + \frac{\lambda_{\max}^3(\mathbf{M})(6+d)^3\alpha^2 l^2}{4(\gamma+t)^2} - \frac{(l\lambda_{\min}(\mathbf{M})-2)v}{2(\gamma+t)^2}.
\end{aligned}
$$

We let $\dfrac{3\sigma^2 l^2\mathrm{tr}(\mathbf{M})}{(\gamma+t)^2} + \dfrac{\lambda_{\max}^2(\mathbf{M})(6+d)^3\alpha^2 l}{8(\gamma+t)} + \dfrac{\lambda_{\max}^3(\mathbf{M})(6+d)^3\alpha^2 l^2}{4(\gamma+t)^2} - \dfrac{(l\lambda_{\min}(\mathbf{M})-2)v}{2(\gamma+t)^2} \leq 0.$
This is equivalent to

$$
6\sigma^2 l^2\mathrm{tr}(\mathbf{M}) + \frac{\lambda_{\max}^3(\mathbf{M})(6+d)^3\alpha^2 l^2}{2} + \frac{\lambda_{\max}^2(\mathbf{M})(6+d)^3\alpha^2 l(\gamma+t)}{4} \leq (l\lambda_{\min}(\mathbf{M})-2)v.
$$

$$
\Rightarrow v \geq \frac{54\mathrm{tr}(\mathbf{M})\sigma^2}{\lambda_{\min}^2(\mathbf{M})} + \frac{9\lambda_{\max}^3(\mathbf{M})(6+d)^3\alpha^2}{2\lambda_{\min}^2(\mathbf{M})} + \frac{3\lambda_{\max}^2(\mathbf{M})(6+d)^3\alpha^2(\gamma+t)}{4\lambda_{\min}(\mathbf{M})}.
$$

$$
\Rightarrow v \geq \frac{54\mathrm{tr}(\mathbf{M})\sigma^2}{\lambda_{\min}^2(\mathbf{M})} + \frac{3\left[6\lambda_{\max}(\mathbf{M}) + 36\mathrm{tr}(\mathbf{M}) + \lambda_{\min}(\mathbf{M})T\right]\lambda_{\max}^2(\mathbf{M})(6+d)^3\alpha^2}{4\lambda_{\min}^2(\mathbf{M})}.
$$

$$
\Rightarrow v \geq \frac{54\mathrm{tr}(\mathbf{M})\sigma^2}{\lambda_{\min}^2(\mathbf{M})} + \frac{3\left[6\lambda_{\max}(\mathbf{M}) + 36\mathrm{tr}(\mathbf{M}) + \lambda_{\min}(\mathbf{M})T\right]\lambda_{\max}^2(\mathbf{M})(6+d)^3\alpha_0^2}{4(T+1)\lambda_{\min}^2(\mathbf{M})}.
$$

$$
\begin{aligned}
\Rightarrow v \geq & \frac{54\mathrm{tr}(\mathbf{M})\sigma^2}{\lambda_{\min}^2(\mathbf{M})} + \frac{3\left[6\lambda_{\max}(\mathbf{M}) + 36\mathrm{tr}(\mathbf{M})\right]\lambda_{\max}^2(\mathbf{M})(6+d)^3\alpha_0^2}{4\lambda_{\min}^2(\mathbf{M})} \\
= & \frac{54\mathrm{tr}(\mathbf{M})\sigma^2}{\lambda_{\min}^2(\mathbf{M})} + Q_1(\alpha_0^2).
\end{aligned}
$$

So, we can finally obtain $v \geq \dfrac{54\mathrm{tr}(\mathbf{M})\sigma^2}{\lambda_{\min}^2(\mathbf{M})} + Q_1(\alpha_0^2).$

Due to the facts

$$
(\gamma+t)^2 \geq (\gamma+t+1)(\gamma+t-1) = (\gamma+t)^2 - 1,
$$

then

$$
\mathbb{E}\left[f(\boldsymbol{x}^{t+1}) - f^*\right] \leq \frac{v}{\gamma+t+1}.
$$

$\square$

### D.3 Proof of Theorem 4.7

If the objective function is not quadratic function, we notice that $\gamma_u \neq 1$ and $\gamma_l \neq 1$. So, we can transform inequality (23) into

$$
\begin{aligned}
f(\boldsymbol{x}^{t+1}) \leq & f(\boldsymbol{x}^t) - \eta \left\langle \nabla f(\boldsymbol{x}^t), \boldsymbol{u}_t \boldsymbol{u}_t^\top \nabla f(\boldsymbol{x}^t, \mathcal{S}_t) + \phi(\boldsymbol{u}_t, \alpha, \boldsymbol{x}^t) \right\rangle \\
& + \frac{\gamma_u \eta^2}{2} \left\| \boldsymbol{u}_t \boldsymbol{u}_t^\top \nabla f(\boldsymbol{x}^t, \mathcal{S}_t) + \phi(\boldsymbol{u}_t, \alpha, \boldsymbol{x}^t) \right\|_{\mathbf{M}(\boldsymbol{z}^t)}^2 .
\end{aligned}
$$

Let us deduce the expectation of $f(\boldsymbol{x}^{t+1})$ for $\boldsymbol{u}$,

$$
\begin{aligned}
\mathbb{E}_{\boldsymbol{u}_t} \left[ f(\boldsymbol{x}^{t+1}) \right] \overset{(20)+(21)}{\leq} & f(\boldsymbol{x}^t) + 3\eta^2 \gamma_u \mathrm{tr}(\mathbf{M}(\boldsymbol{z}^t)) \left\| \nabla f(\boldsymbol{x}^t, \mathcal{S}_t) \right\|^2 + \frac{\eta}{2} \left\| \nabla f(\boldsymbol{x}^t) \right\|^2 \\
& - \eta \left\langle \nabla f(\boldsymbol{x}^t), \nabla f(\boldsymbol{x}^t, \mathcal{S}_t) \right\rangle \\
& + \frac{\left[ 1 + 2\gamma_u \lambda_{\max}(\mathbf{M}(\boldsymbol{z}^t))\eta \right] \lambda_{\max}^2(\mathbf{M}(\boldsymbol{z}^t))(6+d)^3 \gamma_u^2 \alpha^2 \eta}{8} .
\end{aligned}
$$

And we can transform inequality (24) into

$$
\begin{aligned}
\mathbb{E} \left[ f(\boldsymbol{x}^{t+1}) \right] \leq & f(\boldsymbol{x}^t) + 3\eta^2 \sigma^2 \gamma_u \mathrm{tr}(\mathbf{M}(\boldsymbol{z}^t)) \\
& + \frac{\left[ 1 + 2\gamma_u \lambda_{\max}(\mathbf{M}(\boldsymbol{z}^t))\eta \right] \lambda_{\max}^2(\mathbf{M}(\boldsymbol{z}^t))(6+d)^3 \gamma_u^2 \alpha^2 \eta}{8} \\
& - \frac{1}{2} \gamma_l \eta \lambda_{\min}(\mathbf{M}(\boldsymbol{z}^t))(f(\boldsymbol{x}^t) - f^*) .
\end{aligned}
$$

If we let $\mathrm{tr}(\mathbf{M}) = \max_{\boldsymbol{z}^t} \mathrm{tr}(\mathbf{M}(\boldsymbol{z}^t))$, $\lambda_{\min}(\mathbf{M}) = \min_{\boldsymbol{z}^t} \lambda_{\min}(\mathbf{M}(\boldsymbol{z}^t))$ and $\lambda_{\max}(\mathbf{M}) = \max_{\boldsymbol{z}^t} \lambda_{\max}(\mathbf{M}(\boldsymbol{z}^t))$ in the subsequent analysis. Then, we can obtain

$$
\begin{aligned}
& \mathbb{E} \left[ f(\boldsymbol{x}^{t+1}) - f^* - \frac{24\eta \gamma_u \mathrm{tr}(\mathbf{M})\sigma^2 + \left[ 1 + 2\gamma_u \lambda_{\max}(\mathbf{M})\eta \right] \lambda_{\max}^2(\mathbf{M})(6+d)^3 \gamma_u^2 \alpha^2}{4\gamma_l \lambda_{\min}(\mathbf{M})} \right] \\
\leq & \left[ 1 - \frac{1}{2} \eta \gamma_l \lambda_{\min}(\mathbf{M}) \right] \mathbb{E} \left[ f(\boldsymbol{x}^t) - f^* - \frac{24\eta \gamma_u \mathrm{tr}(\mathbf{M})\sigma^2 + \left[ 1 + 2\gamma_u \lambda_{\max}(\mathbf{M})\eta \right] \lambda_{\max}^2(\mathbf{M})(6+d)^3 \gamma_u^2 \alpha^2}{4\gamma_l \lambda_{\min}(\mathbf{M})} \right] \\
\leq & \left[ 1 - \frac{1}{2} \eta \gamma_l \lambda_{\min}(\mathbf{M}) \right]^t \left[ f(x^0) - f^* - \frac{24\eta \gamma_u \mathrm{tr}(\mathbf{M})\sigma^2 + \left[ 1 + 2\gamma_u \lambda_{\max}(\mathbf{M})\eta \right] \lambda_{\max}^2(\mathbf{M})(6+d)^3 \gamma_u^2 \alpha^2}{4\gamma_l \lambda_{\min}(\mathbf{M})} \right] \\
\leq & \left[ 1 - \frac{1}{2} \eta \gamma_l \lambda_{\min}(\mathbf{M}) \right]^t \left[ f(x^0) - f^* \right] .
\end{aligned}
$$

Thus, we can obtain that

$$
\begin{aligned}
\mathbb{E} \left[ f(\boldsymbol{x}^{t+1}) - f^* \right] \leq & \frac{24\eta \gamma_u \mathrm{tr}(\mathbf{M})\sigma^2 + \left[ 1 + 2\gamma_u \lambda_{\max}(\mathbf{M})\eta \right] \lambda_{\max}^2(\mathbf{M})(6+d)^3 \gamma_u^2 \alpha^2}{4\gamma_l \lambda_{\min}(\mathbf{M})} \\
& + \left[ 1 - \frac{1}{2} \eta \gamma_l \lambda_{\min}(\mathbf{M}) \right]^t \left[ f(x^0) - f^* \right] .
\end{aligned}
$$

Let $\sigma = 0$ and a sufficiently small $\alpha$ is chosen, similar to the proof process of E.1, we can obtain the iteration complexity

$$
t = \mathcal{O} \left( \frac{\gamma_u \mathrm{tr}(\mathbf{M})}{\gamma_l \lambda_{\min}(\mathbf{M})} \log \frac{1}{\varepsilon} \right) . \tag{25}
$$

### D.4 Proof of Theorem 4.8

Firstly, if we choose decreasing step size $\eta_t$, we can obtain the following formula

$$
\begin{aligned}
\mathbb{E} \left[ f(\boldsymbol{x}^{t+1}) \right] \leq & f(\boldsymbol{x}^t) + 3\eta_t^2 \sigma^2 \gamma_u \mathrm{tr}(\mathbf{M}(\boldsymbol{z}^t)) \\
& + \frac{\left[ 1 + 2\gamma_u \lambda_{\max}(\mathbf{M}(\boldsymbol{z}^t))\eta_t \right] \lambda_{\max}^3(\mathbf{M}(\boldsymbol{z}^t))(6+d)^3 \gamma_u^2 \alpha^2 \eta_t}{8} \\
& - \frac{1}{2} \gamma_l \eta_t \lambda_{\min}(\mathbf{M}(\boldsymbol{z}^t))(f(\boldsymbol{x}^t) - f^*) .
\end{aligned}
$$

We let $\mathrm{tr}(\mathbf{M}) = \max_{\boldsymbol{z}^t} \mathrm{tr}(\mathbf{M}(\boldsymbol{z}^t))$, $\lambda_{\min}(\mathbf{M}) = \min_{\boldsymbol{z}^t} \lambda_{\min}(\mathbf{M}(\boldsymbol{z}^t))$ and $\lambda_{\max}(\mathbf{M}) = \max_{\boldsymbol{z}^t} \lambda_{\max}(\mathbf{M}(\boldsymbol{z}^t))$ in the subsequent analysis. Then, we need to add $\gamma_u$ and $\gamma_l$ to the appropriate position in the proof process of D.2 like the similar ways we operated in D.3. Suppose that holds for $t > 0$, then

$$
\begin{aligned}
\mathbb{E}\left[f(\boldsymbol{x}^{t+1}) - f^*\right] \leq &3\eta_t^2\sigma^2\gamma_u\mathrm{tr}(\mathbf{M}) + \frac{\left[1 + 2\gamma_u\lambda_{\max}(\mathbf{M})\eta_t\right]\lambda_{\max}^2(\mathbf{M})(6+d)^3\gamma_u^2\alpha^2\eta_t}{8} \\
&+ \left[1 - \frac{1}{2}\gamma_l\eta_t\lambda_{\min}(\mathbf{M})\right]\mathbb{E}\left[f(\boldsymbol{x}^t) - f^*\right] \\
\leq &3\eta_t^2\sigma^2\gamma_u\mathrm{tr}(\mathbf{M}) + \frac{\left[1 + 2\gamma_u\lambda_{\max}(\mathbf{M})\eta_t\right]\lambda_{\max}^2(\mathbf{M})(6+d)^3\gamma_u^2\alpha^2\eta_t}{8} \\
&+ \left[1 - \frac{1}{2}\gamma_l\eta_t\lambda_{\min}(\mathbf{M})\right]\frac{v}{\gamma + t} \\
= &\frac{(\gamma + t - 1)v}{(\gamma + t)^2} + \frac{3\sigma^2 l^2\gamma_u\mathrm{tr}(\mathbf{M})}{(\gamma + t)^2} + \frac{\lambda_{\max}^2(\mathbf{M})(6+d)^3\gamma_u^2\alpha^2 l}{8(\gamma + t)} \\
&+ \frac{\lambda_{\max}^3(\mathbf{M})(6+d)^3\gamma_u^3\alpha^2 l^2}{4(\gamma + t)^2} - \frac{(\gamma_l l\lambda_{\min}(\mathbf{M}) - 2)v}{2(\gamma + t)^2}.
\end{aligned}
$$

We define $Q_2(\alpha_0^2) = \dfrac{3\left[6\gamma_u\lambda_{\max}(\mathbf{M}) + 36\gamma_u\gamma_l\mathrm{tr}(\mathbf{M})\right]\lambda_{\max}^2(\mathbf{M})(6+d)^3\gamma_u^2\alpha_0^2}{4\gamma_l^2\lambda_{\min}^2(\mathbf{M})}.$

Then, we can obtain $v \geq \dfrac{54\gamma_u\mathrm{tr}(\mathbf{M})\sigma^2}{\gamma_l^2\lambda_{\min}^2(\mathbf{M})} + Q_2(\alpha_0^2).$

Finally, we obtain the iteration complexity

$$
t = \mathcal{O}\left(\left[\frac{\gamma_u\mathrm{tr}(\mathbf{M})\sigma^2}{\gamma_l^2\lambda_{\min}^2(\mathbf{M})} + Q_2(\alpha_0^2)\right]\frac{1}{\varepsilon}\right). \tag{26}
$$

# E    PROOF OF MAIN COROLLARIES

In this section, we give some details of proof about Corollary.

## E.1    PROOF OF COROLLARY 4.4

*Proof.* From the proof of Theorem 4.3, if we choose a sufficiently small $\alpha$ in practice, we can find that

$$
\begin{aligned}
\mathbb{E}\left[f(\boldsymbol{x}^{t+1}) - f^*\right] \leq &\frac{6\eta\mathrm{tr}(\mathbf{M})\sigma^2}{\lambda_{\min}(\mathbf{M})} + \left[1 - \frac{1}{2}\eta\lambda_{\min}(\mathbf{M})\right]^t\left[f(x^0) - f^*\right] \\
\overset{\sigma=0}{=} &\left[1 - \frac{1}{2}\eta\lambda_{\min}(\mathbf{M})\right]^t\left[f(\boldsymbol{x}^0) - f^*\right] \\
\leq &\exp\left(-\frac{1}{2}\eta\lambda_{\min}(\mathbf{M})t\right)\left[f(\boldsymbol{x}^0) - f^*\right] \\
\leq &\exp\left(-\frac{\lambda_{\min}(\mathbf{M})}{24\mathrm{tr}(\mathbf{M})}t\right)\left[f(\boldsymbol{x}^0) - f^*\right].
\end{aligned}
$$

Thus, in order to achieve $\varepsilon$-suboptimal solution, $t$ is required to be

$$
\begin{aligned}
t = &\frac{24\mathrm{tr}(\mathbf{M})}{\lambda_{\min}(\mathbf{M})}\left(\log\frac{1}{\varepsilon} + \log\left(f(\boldsymbol{x}^0) - f^*\right)\right) \\
= &\mathcal{O}\left(\frac{\mathrm{tr}(\mathbf{M})}{\lambda_{\min}(\mathbf{M})}\log\frac{1}{\varepsilon}\right).
\end{aligned}
$$

$\square$

# F    EIGENVALUE SKEWNESS ANALYSIS ON REAL-WORLD DATASETS

We analyze the spectral properties of the Hessian matrices at both the initial points and near the optimal solutions across all three datasets. We observe that the eigenvalue spectra of the Hessians exhibit a rapid decay in all cases. Consistent with the findings of Yue et al. (2023), this indicates that many real-world problems naturally possess rapidly decaying Hessian spectra. This phenomenon fundamentally explains why ZSG-type algorithms—with their inherently weak dimensional dependence—tend to perform well and are widely adopted in practice.

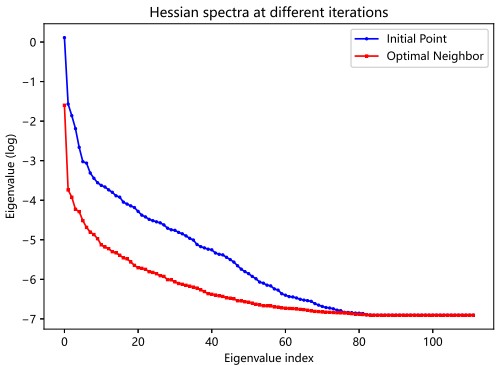

(a) Eigenvalue distribution of the Hessian of 'mushrooms'

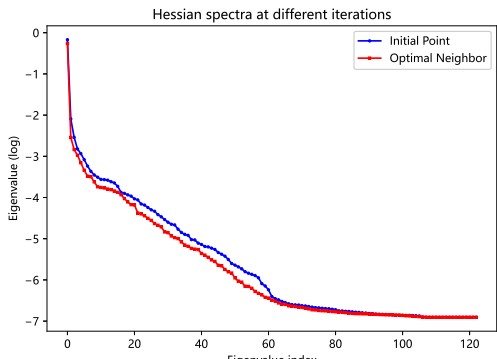

(b) Eigenvalue distribution of the Hessian of 'a8a'

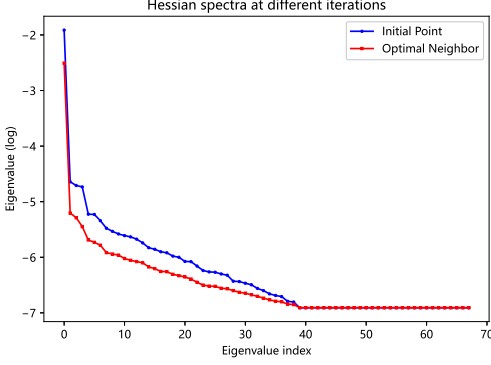

(c) Eigenvalue distribution of the Hessian of 'phishing'

Figure 3: Eigenvalue distributions of the Hessian matrices at the initial points and near the optimal solutions for three logistic regression datasets.

# G    SENSITIVITY ANALYSIS OF $\alpha$ IN LOGISTIC REGRESSION

In the quadratic objective setting, the $\alpha$-related terms do not introduce additional bias into the zeroth-order gradient estimator. Therefore, in this section we analyze the sensitivity of $\alpha$ in the Logistic Regression tasks. For each of the three datasets, we conduct ZSG experiments with $\alpha = 10^0$, $10^{-0.5}$, $10^{-1}$, $10^{-2}$, and $10^{-6}$. The results show that when $\alpha = 10^0$, the large noise markedly impedes convergence across all datasets. When $\alpha = 10^{-0.5}$, the noise has a mild negative effect on convergence for 'mushrooms' and 'a8a'. In addition, we find that $\alpha = 10^{-2}$ is already sufficiently small for most practical problems, indicating that the noise induced by $\alpha$ is easily controlled.

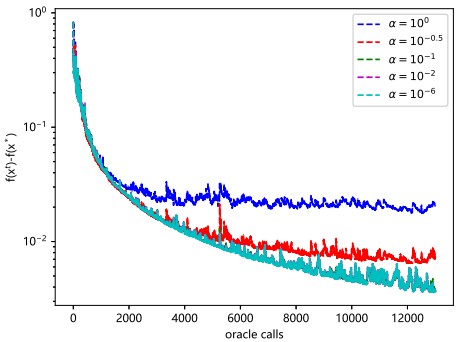

(a) Sensitivity analysis on 'mushrooms'

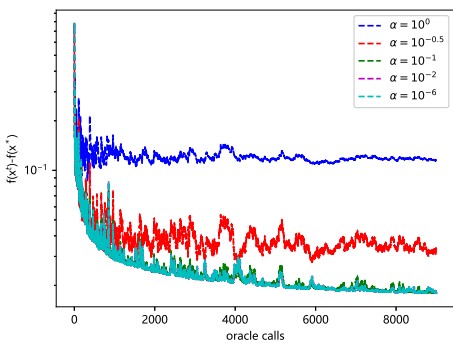

(b) Sensitivity analysis on 'a8a'

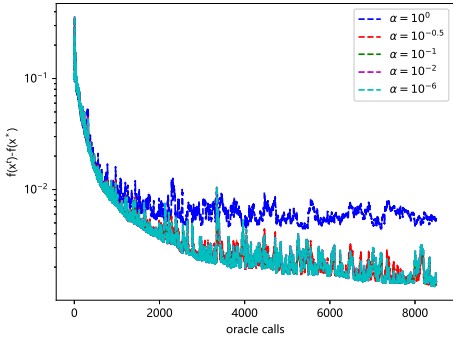

(c) Sensitivity analysis on 'phishing'

Figure 4: The effect of different values of $\alpha$ on model training across three logistic-regression datasets.

# H Algorithm Description of ZSC

---

**Algorithm 2** ZSC: ZO-SGD-Coordinate Method

---

**Input and Initialize:** parameters $\boldsymbol{x} \in \mathbb{R}^d$, loss function $f : \mathbb{R}^d \to \mathbb{R}$, step budget $t$, step size $\eta_t > 0$, perturbation scale $\alpha$, sample distribution $\mathcal{D}$, initial point $\boldsymbol{x}^0 \in \mathbb{R}^d$

**for** $t = 0, 1, \cdots$ **do**

    Sample $\mathcal{S}_t \sim \mathcal{D}$ and $\boldsymbol{e}_t \sim \mathcal{U}^d$

    Query the zeroth-order oracle $f_+^t = f(\boldsymbol{x}^t + \alpha \boldsymbol{e}_t, \mathcal{S}_t)$

    Query the zeroth-order oracle $f_-^t = f(\boldsymbol{x}^t - \alpha \boldsymbol{e}_t, \mathcal{S}_t)$

    Estimating the gradient $\tilde{\nabla} f(\boldsymbol{x}^t, \mathcal{S}_t) = \frac{(f_+^t - f_-^t)}{2\alpha} \cdot \boldsymbol{e}_t$

    $\boldsymbol{x}^{t+1} = \boldsymbol{x}^t - \eta_t \tilde{\nabla} f(\boldsymbol{x}^t, \mathcal{S}_t)$

**end for**

---

