# OpenReview forum: "Stochastic Gaussian Zeroth-Order Optimization: Improved Convergence Analysis under Skewed Hessian Spectra"
_ICLR.cc/2026/Conference — Submitted to ICLR 2026_

### Official Review · Reviewer_XZDu · 2025-10-30

**Soundness:** 3
**Presentation:** 1
**Contribution:** 2
**Rating:** 2
**Confidence:** 3

**Summary:**

The paper studies the zero-order SGD with gaussian gradient estimation. The authors refine the convergence rates, showing that convergence depends on $\text{tr}(\nabla^2 f)$, instead of $\lambda_{\max}(\nabla^2 f)$. They compare the performance of the analyzed algorithm on quadratic functions, as well as on logistic regression on LibSVM datasets.

**Strengths:**

1)The authors improve the existing rates for zero-order optimization, that benefits the skewed Hessian spectra, when $\text{tr}(\nabla^2f) \leq d\lambda_{max}(\nabla^2 f)$.

**Weaknesses:**

1)The authors claim that the terms $P_1(\alpha)$ and $Q_1(\alpha)$ are negligible with the small choice of $\alpha$. However, these terms contain multipliers $\lambda_{\max}^2(\nabla^2 f)d^3$ and $\lambda_{\max}^2(\nabla^2 f)d^3T$, which are frequently large.

2)Unclear writing -- no description of ZSC was given, though, the authors compare the obtained results with it throughout the paper, differences between Theorems 4.5, 4.7, 4.8 are hard to distinguish.

3)Considering $\text{tr}(\nabla^2 f)$ as $\max_{z^t}\nabla^2 f(z^t)$ wuth similar definitions for $\lambda_{\min}(\nabla^2 f)$ and $\lambda_{\max}(\nabla^2 f)$ is a rough estimate. With this analysis, most convex problems might be considered as strongly convex, when $\lambda_{\min} > 0$.

4)The plots do not contain confidence intervals; however, stochastic methods are considered. Also, more complex setups than LibSVM datasets are missing.

**Questions:**

1)The derived stepsize depends on $\text{tr}(\nabla^2 f)$ and $\lambda_{\min}(\nabla^2 f)$. How are they obtained in practice?

2)If we access the objective's Hessian during the training process, why do we consider the derivative-free optimization at the first place?

3)Do we demand gaussian distributions in the scheme proposed in Algorithm 1? What if we consider arbitrary (a)symmetric distribution?

4)Does corollary 4.6 result in sublinear convergence for strongly convex functions? According to Assumption 2.1  $\gamma_l > 0$.

**Details Of Ethics Concerns:**

No additional ethical concerns.

---

> ### Author Response · Authors · 2025-11-22
> **Response to Reviewer XZDu (one)**
>
> Thank you for your feedback. We greatly appreciate your time and effort in evaluating our work.
>
> Q1:The authors claim that the terms $P_{1}(\alpha)$ and $Q_{1}(\alpha)$ are negligible with the small choice of $\alpha$. However, these terms contain multipliers $\lambda_{\max}^{2}(\nabla^{2} f)\, d^{3}$ and $\lambda_{\max}^{2}(\nabla^{2} f)\, d^{3} T$, which are frequently large.
>
> A1:It is possible that our wording led to some misunderstanding. We would like to clarify that, in the zeroth-order optimization literature, it is standard to assume that $\alpha \to 0$. For example, Spall (1992) explicitly requires
> $\alpha_t \to 0$ as $t \to \infty$ and designs schedules of the form $\alpha_t=\frac{\alpha}{t^k}$. Polyak (1987) establishes convergence results under the condition $\alpha_t \to 0$. In the theory developed by Nesterov \& Spokoiny (2017), the $\alpha$ is chosen to be sufficiently small and is required to satisfy $\alpha \le\frac{5}{3(d+4)} \sqrt{\frac{\epsilon}{2L} \cdot \frac{\tau}{L}}$, where
> $\epsilon$ denotes the target accuracy, $L$ is the smoothness constant, and $\tau$ is the strong convexity parameter. The reason that $P_{1}(\alpha)$ and $Q_{1}(\alpha)$ appear in our theoretical analysis is that we explicitly retain the full bias term $\phi( u , \alpha,  x )$ arising from the zeroth-order gradient estimator. Both Malladi et al. (2023) and Zhao et al. (2024) derive their theoretical results under the approximation $\hat{\nabla} f( x ,\mathcal{S}) \approx  u  u ^\top \nabla f( x ,\mathcal{S})$, effectively ignoring the bias term $\phi( u , \alpha,  x )$. Zhao et al. (2024) retain only a qualitative $\mathcal{O}\left(\alpha^2\right)$ term in their analysis. To ensure that our theoretical contribution is more rigorous and reliable, we choose to keep the full bias term $\phi( u , \alpha,  x )$ throughout our analysis. From a theoretical perspective, $\alpha$ dominates both $P_{1}(\alpha)$ and $Q_{1}(\alpha)$. Therefore, as $\alpha \to 0$, both $P_{1}(\alpha)$ and $Q_{1}(\alpha)$ should also vanish.
> In experiments, above works—as well as many other studies in the zeroth-order optimization literature—also choose a sufficiently small $\alpha$, consistent with our work. To present our contribution more clearly and avoid potential misunderstandings, we choose $\alpha\le \sqrt{\frac{\alpha_0}{T+1}}$, which is similar to the requirement $\alpha \propto \sqrt{\epsilon}$ in Nesterov and Spokoiny (2017). We have revised the statements of Theorems 4.5 and 4.8 in the paper.
>
> Theorems 4.5: Let $f$ be a quadratic function, and suppose that Assumption 2.1 and Assumption 2.2 hold. Suppose the update rule of $ x ^{t+1}$ follows Eq. 7. Assume the step size follows the decreasing form $ \eta_t = \frac{l}{\gamma + t}$, where $\gamma >0$ and the intermediate parameter $l=\frac{3}{\lambda_{\min}(\mathbf{M} )}$. Let $t_{\max} = T$ and $\alpha\le \sqrt{\frac{\alpha_0}{T+1}}$, where $\alpha_0$ is a tunable perturbation scale. We define $Q_1(\alpha_0^2)=\frac{3\left[6\lambda _{\max}(\mathbf{M})+36\mathrm{tr}(\mathbf{M} )\right]\lambda^2 _{\max}(\mathbf{M})(6+d)^3\alpha_0^2}{4\lambda^2 _{\min}(\mathbf{M} )}$. The initial step size satisfies $\eta _0=\frac{l}{\gamma }\le \frac{1}{12\mathrm{tr}(\mathbf{M})}$, which implies $\gamma  \geq \frac{36\mathrm{tr}(\mathbf{M})}{\lambda _{\min}(\mathbf{M})}$. Next, define the auxiliary parameter  $v  =\max \\{ \gamma (f( x ^0)-f^{\*}),\frac{54\mathrm{tr}(\mathbf{M})\sigma ^2}{\lambda^2 _{\min}(\mathbf{M})} +Q_1(\alpha_0^2) \\}$. Then, for every integer $t$ with $0 \le t \le T$, we can obtain the following result
> $$\mathbb{E}\left[f( x ^{t})-f^{\*}\right]\leq \frac{v  }{\gamma +t}.$$

---

> ### Author Response · Authors · 2025-11-22
> **Response to Reviewer XZDu (two)**
>
> A1:
> Theorems 4.8: Suppose $f$ is in the general form described in problem 1 and assumption 2.1,2.2 hold. Suppose the update rule of $ x ^{t+1}$ follows Eq. 7. Assume the step size follows the decreasing form $ \eta_t = \frac{l}{\gamma + t}$, where $\gamma >0$ and the intermediate parameter $l=\frac{3}{\gamma_l\lambda_{\min}(\mathbf{M} )}$. Let $t_{\max} = T$ and $\alpha\le \sqrt{\frac{\alpha_0}{T+1}}$, where $\alpha_0$ is a tunable perturbation scale. We define $Q_2(\alpha_0^2)=\frac{3\left[6\gamma_u\lambda _{\max}(\mathbf{M})+36\gamma_u\gamma_l\mathrm{tr}(\mathbf{M} )\right]\lambda^2 _{\max}(\mathbf{M})(6+d)^3\gamma_u^2\alpha_0^2}{4\gamma_l^2\lambda^2 _{\min}(\mathbf{M} )}$. The initial step size satisfies $\eta _0=\frac{l}{\gamma }\le \frac{1}{12\gamma_u\mathrm{tr}(\mathbf{M})}$, which implies $\gamma  \geq \frac{36\gamma_u\mathrm{tr}(\mathbf{M})}{\lambda _{\min}(\mathbf{M})}$. Next, define the auxiliary parameter  $v  =\max\\{ \gamma (f( x ^0)-f^{\*}),\frac{54\gamma_u\mathrm{tr}(\mathbf{M})\sigma ^2}{\gamma_l^2\lambda^2 _{\min}(\mathbf{M})} +Q_2(\alpha_0^2) \\}$.
>
> We define $\mathrm{tr}(\mathbf{M})=\max_{  z ^t}\mathrm{tr}(\mathbf{M}(  z ^t))$, with similar definitions for $\lambda _{\min}(\mathbf{M})$ and $\lambda _{\max}(\mathbf{M})$. Then, for every integer $t$ with $0 \le t \le T$, we can obtain the following result
> \begin{align*}
> 	\mathbb{E}\left[f( x ^{t})-f^{\*}\right]\leq \frac{v  }{\gamma +t}.
> \end{align*}
> Q2:Unclear writing -- no description of ZSC was given, though, the authors compare the obtained results with it throughout the paper, differences between Theorems 4.5, 4.7, 4.8 are hard to distinguish.
>
> A2:You may be referring to the fact that ZSC does not have an algorithmic flowchart corresponding to ZSG, such as Algorithm 1 in the paper. Due to space limitations, we have placed the algorithmic flowchart for ZSC in Section H. However, this does not affect the understanding of the main content or the theoretical contributions of the paper. As stated on lines 289 and 358, we explain that “To obtain the ZSC algorithm, we substitute $  u _t \sim \mathcal{N}(\mathbf{0}, \mathbf{I}_d) $ with $ e _t \sim \mathcal{U}^d $, where $\mathcal{U}^d$ denotes the uniform distribution over the set of standard basis vectors in $\mathbb{R}^d$”. In other words, the only difference between ZSC and ZSG is replacing $  u _t $ in Algorithm 1 with $  e _t $. The rest of the algorithmic procedure remains identical. Due to space constraints, we did not include a separate flowchart for ZSC. Theorem 4.5 is established specifically for quadratic functions. Studying the quadratic case is often straightforward and intuitive, and this approach is widely adopted in the zeroth-order optimization literature, e.g., Yue et al. (2023) and Wang et al. (2024). Since we employ the quadratic regularity assumption, our analysis naturally generalizes beyond quadratic objectives, and Theorems 4.7 and 4.8 indeed demonstrate that our theoretical contributions are not limited to the quadratic setting. In particular, Theorem 4.7 further illustrates the comprehensiveness of our analytical framework by building upon Corollary 4.4, as it generalizes the theoretical contribution of Wang et al. (2024).

---

> ### Author Response · Authors · 2025-11-22
> **Response to Reviewer XZDu (three)**
>
> Q3:Considering $\mathrm{tr}(\nabla^{2} f)$ as $\max_{z^t} \nabla^{2} f(z^{t})$ with similar definitions for $\lambda_{\min}(\nabla^{2} f)$ and $\lambda_{\max}(\nabla^{2} f)$ is a rough estimate. With this analysis, most convex problems might be considered as strongly convex, when $\lambda_{\min} > 0$.
>
> A3:We indeed make use of strong convexity in our derivations, and we explicitly state in line 144 of the paper that $\mathbf{M} $ is assumed to be positive definite. Although Frangella et al. (2023) introduce the quadratic regularity assumption, the objective function they study is $F( w ) := \frac{1}{n} \sum_{i=1}^{n} f_i( w ) + \frac{\nu}{2} \| w \|^{2}$, which is itself strongly convex. The key role of the quadratic regularity assumption is that it extends smoothness and strong convexity conditions to the Hessian norm. This extension enables us to exploit Hessian information in our analysis and thereby reveal the theoretical superiority of ZSG. To make our analysis more comprehensive and reliable, we additionally provide a convergence analysis of ZSG in the non-convex setting. For clarity, we temporarily ignore the term $\phi( u , \alpha,  x )$, omit the quadratic lower-bound inequality, and make an initial assumption that $\eta_t\le \frac{1}{3\gamma_u\mathrm{tr}(\mathbf{M})}$. Here, $\mathbf{M}$ is no longer required to be the Hessian matrix, rather, this follows the standard $\mathbf{M}$-smoothness assumption ($L$–smoothness assumption is a special case of $\mathbf{M}$–smoothness for $\gamma_u\mathbf{M}=L\mathbf{I}$). The main steps of ZSG are as follows:
>
> \begin{align*}
> 	f( x ^{t+1})
> 	\le&
> 	f( x ^t) -\eta_t\left\langle \nabla f( x ^t),  u _t u _t^\top\nabla f( x ^t,\mathcal{S}_t)\right\rangle +\frac{\gamma_u\eta_t^2}{2}\lVert  u _t u _t^\top\nabla f( x ^t,\mathcal{S}_t)\rVert _{\mathbf{M} }^2.
> \end{align*}
>  \begin{align*}
> 	\mathbb{E}\left[f( x ^{t+1})\right]
> 	\le& f( x ^t)-\eta_t\mathbb{E}\left[\lVert \nabla f( x ^t)\rVert ^2\right]+\frac{3\gamma_u\eta_t^2\mathrm{tr}(\mathbf{M})}{2}\mathbb{E}\left[\lVert \nabla f( x ^t)\rVert ^2\right]+\frac{3\gamma_u\eta_t^2\mathrm{tr}(\mathbf{M})\sigma^2}{2}.
> \end{align*}
>  \begin{align*}
> 	\mathbb{E}\left[f( x ^{t+1})\right]
> 	\le& f( x ^t)-\eta_t(1-\frac{3\gamma_u\mathrm{tr}(\mathbf{M})}{2}\eta_t)\mathbb{E}\left[\lVert \nabla f( x ^t)\rVert ^2\right]+\frac{3\gamma_u\eta_t^2\mathrm{tr}(\mathbf{M})\sigma^2}{2}.
> \end{align*}
>  \begin{align*}
> 	\mathbb{E}\left[f( x ^{t+1})\right]
> 	\le&f( x ^t)-\frac{\eta_t}{2}\mathbb{E}\left[\lVert \nabla f( x ^t)\rVert ^2\right]+\frac{3\gamma_u\eta_t^2\mathrm{tr}(\mathbf{M})\sigma^2}{2}.
> \end{align*}
> Transpose the terms to get
>  \begin{align*}
>       \frac{\eta_t}{2}\mathbb{E}\left[\lVert \nabla f( x ^t)\rVert^2\right]
>       \le
>       \mathbb{E}\left[f( x ^t)-f( x ^{t+1})\right]
>       + \frac{3\gamma_t \mathrm{tr}(\mathbf{M})\sigma^2 \eta_t^2}{2}.
> \end{align*}
>
>  \begin{align*}
>       \frac{\sum_{t=0}^{T-1} \eta_t\mathbb{E}\left[\lVert\nabla f(x^t)\rVert^2\right]}{2}
>       \le
>       \sum_{t=0}^{T-1} \mathbb{E}\left[f( x ^t)-f( x ^{t+1})\right] + \frac{3\gamma_u \mathrm{tr}(\mathbf{M})\sigma^2\sum_{t=0}^{T-
>      1}\eta_t^2}{2}.
> \end{align*}
>
>  \begin{align*}
>       \frac{\sum_{t=0}^{T-1} \eta_t\mathbb{E}\left[\lVert\nabla f(x^t)\rVert^2\right]}{2}
>       \le
>       (f( x ^0)-f^{\*}) + \frac{3\gamma_u \mathrm{tr}(\mathbf{M})\sigma^2\sum_{t=0}^{T-1}\eta_t^2}{2}.
> \end{align*}
> So, we can obtain
>
> \begin{align*}
>     \min_{0\le t \le T-1}
>     \mathbb{E}\left[\lVert\nabla f( x ^t)\rVert^2\right]
>     \le
>     \frac{2(f(x^0)-f^{\*}) + 3\gamma_u\mathrm{tr}(\mathbf{M})\sigma^2 \sum_{t=0}^{T-1} \eta_t^2}{\sum_{t=0}^{T-1} \eta_t}.
> \end{align*}
> Furthermore, if we choose $\eta_t=\eta=\min \\{ \frac{1}{3 \gamma_u \mathrm{tr}(\mathbf{M})},\sqrt{\frac{2(f( x ^0)-f^{\*})}{3 \gamma_u \mathrm{tr}(\mathbf{M})\sigma^2 T}} \\}$, we can obtain
> \begin{align*}
>     \min_{0\le t \le T-1}
>     \mathbb{E}\left[\lVert\nabla f( x ^t)\rVert^2\right]=\mathcal{O}(\sqrt{\frac{\gamma_u \mathrm{tr}(\mathbf{M})\sigma^2 (f( x ^0)-f^{\*})}{T}}).
> \end{align*}
> By the same reasoning, a convergence analysis can also be established for ZSC:
> \begin{align*}
>     \min_{0\le t \le T-1}
>     \mathbb{E}\left[\lVert\nabla f( x ^t)\rVert^2\right]
>     =\mathcal{O}(\sqrt{\frac{\gamma_u d\lambda _{\max}(\mathbf{M})\sigma^2 (f( x ^0)-f^{\*})}{T}}).
> \end{align*}
> Therefore, our theoretical insights continue to hold in the non-convex setting. Due to space limitations, we are unable to include the full analysis in the current version of the paper. If deemed necessary, we can incorporate a discussion of the non-convex case in the camera-ready version.

---

> ### Author Response · Authors · 2025-11-22
> **Response to Reviewer XZDu (four)**
>
> Q4:The plots do not contain confidence intervals; however, stochastic methods are considered. Also, more complex setups than LibSVM datasets are missing.
>
> A4:While incorporating confidence intervals into the experimental results can certainly be beneficial, the experiments we provide are sufficient to validate our theoretical findings. The ZSG algorithm has already been widely adopted in the research community, and our contribution does not lie in conducting additional empirical studies. Rather, our focus is on providing a theoretical explanation for why ZSG exhibits superior performance in practice, thereby filling an existing gap in the current literature. Numerous studies have already employed ZSG for fine-tuning large language models, while ZSC is typically abandoned in practice such as (Malladi et al., 2023), (Zhao et al., 2024), (Zhang et al., 2024), (Chen et al., 2024), and so on. Moreover, several other works do not report confidence intervals, for example (Wu et al., 2024), (Sinha et al., 2024), (Ferbach et al., 2024) and (Shui et al., 2024).
>
> Q5:The derived stepsize depends on $\mathrm{tr}(\nabla^{2} f)$ and $\lambda_{\min}(\nabla^{2} f)$. How are they obtained in practice?
>
> A5:We would like to clarify that our contribution lies in theoretically uncovering the reason behind the superiority of ZSG. When information related to the step-size is difficult to obtain in practice, a common strategy is to start with a relatively small step-size and then adjust it progressively. Rakhlin et al. (2011) have established the optimal convergence rate $\mathbb{E}\!\left[F(\mathbf{w}_T)-F(\mathbf{w}^*)\right] \le \frac{2\mu G^2}{\lambda^2 T}$, along with the recommended step-size $\eta_t = \frac{1}{\lambda t}$ for SGD, where $\lambda$ denotes the strong convexity parameter. Their work also focuses on the theoretical aspects rather than practical step-size tuning strategies. Malladi et al. (2023) and Zhao et al. (2024) also employ zeroth-order optimization methods for fine-tuning LLMs and attempt to provide convergence analyses. Similarly, they rely on theoretically appropriate step-sizes $\eta_t$ when deriving their convergence guarantees, while in practice they simply select several small, commonly used learning rates for the experiments. In summary, the convergence analysis primarily serves as a theoretical contribution, whereas the choice and tuning of learning-rate schedules are engineering considerations. Our work focuses on addressing the theoretical gap underlying the superior performance of ZSG.
>
> Q6:If we access the objective's Hessian during the training process, why do we consider the derivative-free optimization at the first place?
>
> A6:First, as explained in A5, the Hessian information is used solely to establish theoretical convergence bounds for ZSG and ZSC; whether one can obtain its exact values in practice does not affect our theoretical contribution. Second, regarding learning-rate tuning, a common practical strategy is to start with a relatively small learning rate and then adjust it progressively. Finally, if one insists on obtaining Hessian information during training, the standard approach is not to compute the full Hessian matrix but to estimate its diagonal using zeroth-order methods. For example, Zhao et al. (2024) estimate the diagonal Hessian $\Sigma_t'(\theta)$ via $\Sigma_t'( x )= \frac{1}{2} \left[ \frac{\Delta f_t}{\alpha^{2}} \cdot \left( \Sigma_t^{-1/2} u u^{\top} \Sigma_t^{-1/2} - \Sigma^{-1}\right) \right]$, where $\Delta f_t=f( x ^t+\alpha\Sigma^{1/2}  u ,\mathcal{S}_t)+f( x ^t-\alpha\Sigma^{1/2}  u ,\mathcal{S}_t)-2f( x ^t,\mathcal{S}_t)$, and one may initialize with $\Sigma_0 = \mathbf{I}_d$. Moreover, Zhao et al. (2024) apply an exponential moving average (EMA) to denoise the diagonal Hessian estimates:
>
> $\Sigma_{t+1}^{-1}= (1 - \mu_t)\Sigma_t^{-1} + \mu_t \|diag(\Sigma_t')\|.$

---

> ### Author Response · Authors · 2025-11-22
> **Response to Reviewer XZDu (five)**
>
> Q7:Do we demand gaussian distributions in the scheme proposed in Algorithm 1? What if we consider arbitrary (a)symmetric distribution?
>
> A7:Gaussian distribution is merely the most common choice. Alternative distributions may also be used, provided that they satisfy certain required conditions. Recently, Ye et al. (2025) introduce the concept of Oblivious Randomized Sketching within their zeroth-order optimization analysis framework, and provide theoretical conditions that the distribution of the direction vectors must satisfy. As an illustrative example, consider the quadratic case. They define the approximate gradient in the following form: $\mathbf{g}(\mathbf{x}) = S S^{\top} \nabla \phi(\mathbf{x}) + S \mathbf{v}$ and $\lVert \mathbf{v}\rVert ^{2} \le \frac{\ell \sigma^{2}}{\alpha^{2}}$, where $\mathbf{v}$ is an $l$-dimension vector whose $i$-th entry $v^{(i)} = \frac{\zeta(\mathbf{x} + \alpha \mathbf{s}^{(i)}) - \zeta(\mathbf{x} - \alpha \mathbf{s}^{(i)})}{2\alpha}$ and $\mathbf{s}^{(i)}$ being the $i$-th column of $S$. Through $\phi(\mathbf{x}_{t+1})= \phi(\mathbf{x}_t) - \frac{\eta}{2} \lVert  S_t^{\top} \nabla \phi(\mathbf{x}_t) \rVert ^{2} + \frac{\eta}{2} \lVert  \mathbf{v}_t \rVert ^{2} + \eta^{2} \lVert  S_t S_t^{\top} \nabla \phi(\mathbf{x}_t) \rVert _{A}^{2} + \eta^{2} \lVert  S_t \mathbf{v}_t \rVert _{A}^{2}$, they observe that, in order for the descent direction $\mathbf{g}(\mathbf{x}_t)$ to be effective, the higher-order terms $\eta^{2} \lVert  S_t S_t^{\top} \nabla \phi(\mathbf{x}_t) \rVert _{A}^{2}$ and $\eta^{2} \lVert  S_t \mathbf{v}_t \rVert _{A}^{2}$ must be controlled. Focusing on the ﬁrst term as an example, $\lVert  S_t S_t^{\top} \nabla \phi(\mathbf{x}_t) \rVert _{A}^{2}\le\lVert  A^{1/2} S_t S_t^{\top} A^{1/2} \rVert _{2}\cdot\lVert S_t^{\top} \nabla \phi(\mathbf{x}_t) \rVert ^{2}
> $. The key quantity in this upper bound is $\lVert  A^{1/2} S_t S_t^{\top} A^{1/2} \rVert _{2}$, which directly affects the stability and convergence behavior of the zeroth-order algorithm. Then, any randomized sketching matrix $S$ that satisfies Definition 1 in Ye et al. (2025) can be used. They further analyze several classes of sketching matrices that satisfy Definition 1, including Gaussian sketching matrices, Rademacher sketching matrices, the Subsampled Randomized Hadamard Transform, and sparse embedding sketching matrices.
>
> Q8:Does corollary 4.6 result in sublinear convergence for strongly convex functions? According to Assumption 2.1 $\gamma_l>0$.
>
> A8:Yes. In addition, we have included in A3 a comparison of the convergence performance of ZSG and ZSC in the non-convex setting.

---

> ### Author Response · Authors · 2025-11-22
> **Response to Reviewer XZDu (six)**
>
> Reference List
>
> [1]J.C. Spall. Multivariate stochastic approximation using a simultaneous perturbation gradient approximation. IEEE Transactions on Automatic Control, 37(3):332–341, 1992. doi: 10.1109/9.119632.
>
> [2]Yurii Nesterov and Vladimir Spokoiny. Random gradient-free minimization of convex functions. Foundations of Computational Mathematics, 17(2):527–566, 2017.
>
> [3]B.Polyak. Introduction to Optimization. Optimization Software - Inc., Publications
> Division, New York (1987).
>
> [4]Sadhika Malladi, Tianyu Gao, Eshaan Nichani, Alex Damian, Jason D Lee, Danqi Chen, and Sanjeev Arora. Fine-tuning language models with just forward passes. Advances in Neural Information Processing Systems, 36:53038–53075, 2023.
>
> [5]Yanjun Zhao, Sizhe Dang, Haishan Ye, Guang Dai, Yi Qian, and Ivor W Tsang. Second-order fine-tuning without pain for llms: A hessian informed zeroth-order optimizer. arXiv preprint arXiv:2402.15173, 2024.
>
> [6]Pengyun Yue, Long Yang, Cong Fang, and Zhouchen Lin. Zeroth-order optimization with weak dimension dependency. In The Thirty Sixth Annual Conference on Learning Theory, pp. 4429–4472. PMLR, 2023.
>
> [7]Yilong Wang, Haishan Ye, Guang Dai, and Ivor Tsang. Can gaussian sketching converge faster on a preconditioned landscape? In Forty-first International Conference on Machine Learning, 2024.
>
> [8]Zachary Frangella, Pratik Rathore, Shipu Zhao, and Madeleine Udell. Promise: Preconditioned stochastic optimization methods by incorporating scalable curvature estimates. arXiv preprint arXiv:2309.02014, 2023.
>
> [9]Zhang, Yihua, Pingzhi Li, Junyuan Hong, Jiaxiang Li, Yimeng Zhang, Wenqing Zheng, Pin-Yu Chen et al. "Revisiting zeroth-order optimization for memory-efficient llm fine-tuning: A benchmark." arXiv preprint arXiv:2402.11592 (2024).
>
> [10]Chen, Yiming, Yuan Zhang, Liyuan Cao, Kun Yuan, and Zaiwen Wen. "Enhancing zeroth-order fine-tuning for language models with low-rank structures." arXiv preprint arXiv:2410.07698 (2024).
>
> [11]Wu, Jianzong, Xiangtai Li, Yanhong Zeng, Jiangning Zhang, Qianyu Zhou, Yining Li, Yunhai Tong, and Kai Chen. "Motionbooth: Motion-aware customized text-to-video generation." Advances in Neural Information Processing Systems 37 (2024): 34322-34348.
>
> [12]Sinha, Abhishek, and Rahul Vaze. "Optimal algorithms for online convex optimization with adversarial constraints." Advances in Neural Information Processing Systems 37 (2024): 41274-41302.
>
> [13]Ferbach, Damien, Quentin Bertrand, Avishek Joey Bose, and Gauthier Gidel. "Self-Consuming Generative Models with Curated Data Provably Optimize Human Preferences." In NeurIPS, pp. 1-27. 2024.
>
> [14]Shui, Zhongyi, Jianpeng Zhang, Weiwei Cao, Sinuo Wang, Ruizhe Guo, Le Lu, Lin Yang et al. "Large-scale and Fine-grained Vision-language Pre-training for Enhanced CT Image Understanding." In The Thirteenth International Conference on Learning Representations.
>
> [15]Alexander Rakhlin, Ohad Shamir, and Karthik Sridharan. Making gradient descent optimal for strongly convex stochastic optimization. arXiv preprint arXiv:1109.5647, 2011.
>
> [16]Ye, Haishan, Xiangyu Chang, and Xi Chen. "A Unified Zeroth-Order Optimization Framework via Oblivious Randomized Sketching." arXiv preprint arXiv:2510.10945 (2025).

---

### Official Review · Reviewer_ivSN · 2025-10-31

**Soundness:** 1
**Presentation:** 1
**Contribution:** 3
**Rating:** 2
**Confidence:** 4

**Summary:**

The paper analyzes **Gaussian-based zeroth-order stochastic gradient descent (ZSG)** and compares it with **coordinate-based zeroth-order SGD (ZSC)**.
It introduces **Assumption 2.1 : Quadratic Regularity (QR)**, a deterministic Hessian-metric condition generalizing smoothness and strong convexity.
Under this assumption the authors derive iteration-complexity bounds suggesting that ZSG enjoys milder dimensional dependence (\(\operatorname{tr}(M)\)) than ZSC (\(d\,\lambda_{\max}(M)\)), particularly for skewed Hessian spectra.
Experiments on synthetic quadratics and logistic regression qualitatively support the idea that Gaussian perturbations help under ill-conditioning.

However, several technical and presentation issues seriously weaken the results.
Most importantly, **Theorems 4.5 and 4.8 mis-state convergence rates**: the quantities \(Q_1\) and \(Q_2\) depend on the total iteration count T, so the bounds do **not** imply convergence to zero for stochastic functions.
Combined with ambiguous assumptions and missing discussion of validity domains, the paper’s theoretical claims are overstated.

**Strengths:**

- Provides a careful deterministic analysis under a clear quadratic-regularity assumption.
- Offers intuition on why Gaussian perturbations can mitigate poor conditioning.
- Experiments qualitatively match the deterministic predictions.

**Weaknesses:**

1. **Mis-stated main theorems (4.5 & 4.8).**
   The “constants” \(Q_1\) and \(Q_2\) depend explicitly on T through cumulative step-size and variance terms.
   This destroys asymptotic convergence: the error bound does not vanish as T → ∞.
   Despite this, the text claims a “sublinear” convergence in the *stochastic* case.
   The presentation conceals the dependence, giving the impression of a stronger result than actually proved.

2. **Failure to handle stochastic functions.**
   Because the variance term grows with T, the analysis effectively applies only to deterministic or finite-sum settings.
   There is no uniform bound on stochastic noise, so the claimed results do **not** establish convergence for genuinely stochastic oracles.
   The theory should have been presented as deterministic analysis rather than stochastic convergence.

3. **Ambiguity in Assumption 2.1.**
   The constants \(\gamma_u,\gamma_l\) are written as if they may depend on x,y,z, which would make the inequalities tautological.
   For the theorems to hold, they must be global constants independent of those points.
   This appears to be a typographical error that needs correction.

4. **Lack of discussion of when assumptions hold.**
   The paper should explicitly identify and justify classes of functions satisfying QR (e.g., quadratics, certain regularized GLMs).
   Beyond the trivial quadratic case, examples are only hinted at and never proved.

5. **Limited novelty and scope.**
   Algorithmically, ZSG is standard (Gaussian SPSA / NES).
   Experiments are small-scale and deterministic; no tests on high-variance or nonconvex settings.

6. **Lack of transparency.**
   By labeling \(Q_1,Q_2\) as constants and not clarifying their T-dependence, the manuscript obscures a fundamental limitation of the analysis.

**Questions:**

1. Can you formally characterize non-quadratic functions (e.g., logistic or least-squares objectives) that satisfy the QR condition with global constants?
2. Would a fully deterministic framing (σ = 0) strengthen the paper?
3. How could the variance term be controlled to extend the results to true stochastic settings?  Maybe use momentum?
4. Can you restate Theorems 4.5 and 4.8 with explicit T-dependence and honest asymptotic interpretation?

---

> ### Author Response · Authors · 2025-11-22
> **Response to Reviewer ivSN (one)**
>
> Thank you for your comments, which give us the opportunity to further improve the clarity and presentation of our work.
>
> Q1:Mis-stated main theorems (4.5 \& 4.8). The “constants” ($Q_1$) and ($Q_2$) depend explicitly on T through cumulative step-size and variance terms. This destroys asymptotic convergence, the error bound does not vanish as $T \to \infty$. Despite this, the text claims a “sublinear” convergence in the stochastic case. The presentation conceals the dependence, giving the impression of a stronger result than actually proved.
>
> A1:Thank you for pointing this out. We apologize for the confusion caused by our writing, which may have led to a misunderstanding of our contribution. First, the terms in $Q_1$ and $Q_2$ that depend on $T$ arise from the bias $\phi( u , \alpha,  x )$ introduced by using zeroth-order methods to approximate gradient information. Both Malladi et al. (2023) and Zhao et al. (2024) derive their theoretical results under the approximation $\hat{\nabla} f( x ,\mathcal{S}) \approx  u  u ^\top \nabla f( x ,\mathcal{S})$, effectively ignoring the bias term $\phi( u , \alpha,  x )$. Zhao et al. (2024) retain only a qualitative $\mathcal{O}\left(\alpha^2\right)$ term in their analysis. To ensure that our theoretical contribution is more rigorous and reliable, we choose to keep the full bias term $\phi( u , \alpha,  x )$ throughout our analysis. From a theoretical standpoint, $Q_1$ and $Q_2$ should not be treated as constants, because $\alpha$ is assumed to approach zero--this is a standard premise in the analysis of zeroth-order optimization algorithms. For example, Spall (1992) explicitly requires
> $\alpha_t \to 0$ as $t \to \infty$ and designs schedules of the form $\alpha_t=\frac{\alpha}{t^k}$. Polyak (1987) establishes convergence results under the condition $\alpha_t \to 0$. In the theory developed by Nesterov \& Spokoiny (2017), the $\alpha$ is chosen to be sufficiently small and is required to satisfy $\alpha \le\frac{5}{3(d+4)} \sqrt{\frac{\epsilon}{2L} \cdot \frac{\tau}{L}}$, where
> $\epsilon$ denotes the target accuracy, $L$ is the smoothness constant, and $\tau$ is the strong convexity parameter. Our intended message is that the $\alpha^2$ term dominates both $Q_1$ and $Q_2$, and therefore, as $\alpha \to 0$, both $Q_1$ and $Q_2$ should also vanish. To make this point clearer and avoid any misunderstanding, we choose $\alpha\le \sqrt{\frac{\alpha_0}{T+1}}$, which is similar to the requirement $\alpha \propto \sqrt{\epsilon}$ in Nesterov and Spokoiny (2017). We have revised the statements of Theorems 4.5 and 4.8 in the paper.
>
> Theorems 4.5: Let $f$ be a quadratic function, and suppose that Assumption 2.1 and Assumption 2.2 hold. Suppose the update rule of $ x ^{t+1}$ follows Eq. 7. Assume the step size follows the decreasing form $ \eta_t = \frac{l}{\gamma + t}$, where $\gamma >0$ and the intermediate parameter $l=\frac{3}{\lambda_{\min}(\mathbf{M} )}$. Let $t_{\max} = T$ and $\alpha\le \sqrt{\frac{\alpha_0}{T+1}}$, where $\alpha_0$ is a tunable perturbation scale. We define $Q_1(\alpha_0^2)=\frac{3\left[6\lambda _{\max}(\mathbf{M})+36\mathrm{tr}(\mathbf{M} )\right]\lambda^2 _{\max}(\mathbf{M})(6+d)^3\alpha_0^2}{4\lambda^2 _{\min}(\mathbf{M} )}$. The initial step size satisfies $\eta _0=\frac{l}{\gamma }\le \frac{1}{12\mathrm{tr}(\mathbf{M})}$, which implies $\gamma  \geq \frac{36\mathrm{tr}(\mathbf{M})}{\lambda _{\min}(\mathbf{M})}$. Next, define the auxiliary parameter  $v  =\max \\{ \gamma (f( x ^0)-f^{\*}),\frac{54\mathrm{tr}(\mathbf{M})\sigma ^2}{\lambda^2 _{\min}(\mathbf{M})} +Q_1(\alpha_0^2) \\}$. Then, for every integer $t$ with $0 \le t \le T$, we can obtain the following result
> $$\mathbb{E}\left[f( x ^{t})-f^{\*}\right]\leq \frac{v  }{\gamma +t}.$$

---

> > ### Comment · Reviewer_ivSN · 2025-11-24
> >
> > Thanks for the answer.
> >
> > You should have done this from the start.
> > Previous works that ignore the bias are simply mistaken and shouldn't be taken as an example. As you rightfully answered, this term is of a smaller order in $\alpha$ that can be dealt with by taking it small enough.

---

> > > ### Comment · Reviewer_ivSN · 2025-11-24
> > > **Clarification about Assumption 2.1**
> > >
> > > In A 2.1, you say that $M$ is the Hessian matrix of $f$, then you use it in the norm, which seems to suggest that you need $f$ to be convex (at least) for the norm to have a meaning. I would appreciate your comments on this, particularly regarding how this assumption would apply to potentially non-convex functions.
> > >
> > > Also, can you relax the assumption to work with $z=x$?

---

> > > > ### Author Response · Authors · 2025-11-25
> > > > **Clarification about Assumption 2.1**
> > > >
> > > > Thank you for your further comments.
> > > >
> > > > Q1:In A 2.1, you say that $\mathbf{M}$ is the Hessian matrix of $f$, then you use it in the norm, which seems to suggest that you need $f$ to be convex (at least) for the norm to have a meaning. I would appreciate your comments on this, particularly regarding how this assumption would apply to potentially non-convex functions.
> > > >
> > > > A1:For the non-convex setting, we only rely on the upper quadratic regularity inequality. Importantly, this condition does not require $\mathbf{M}(z)$ to be the Hessian. In fact, our proofs only need the existence of a fixed positive definite matrix $\mathbf{M}$ such that
> > > > $$
> > > > f(y) \le f(x)+ \left\langle\nabla f(x), y-x \right\rangle + \frac{\gamma_u}{2}\lVert y- x\rVert_{\mathbf{M}}^2.
> > > > $$
> > > > This is analogous to the standard $L$-smoothness assumption ($L$–smoothness assumption is a special case of $\mathbf{M}$–smoothness for $\gamma_u\mathbf{M}=L\mathbf{I}$) commonly used in non-convex optimization. Thus, no convexity or strong convexity is implicitly assumed in the non-convex part of the analysis we provided above. So, we successfully show that ZSG still enjoys a superior complexity guarantee in the non-convex setting. When adding the proof for the non-convex case in the camera-ready version, we will clarify the differences in the assumptions.
> > > >
> > > > Q2:Can you relax the assumption to work with $z=x$?
> > > >
> > > > A2:By taking $z=x$, the quadratic regularity assumption can be relaxed to its two-point simplified form without affecting our conclusions.

---

> > > ### Author Response · Authors · 2025-11-25
> > >
> > > Thank you for your constructive comments.
> > >
> > > In the field of zeroth-order optimization, it is standard to adopt a sufficiently small perturbation parameter. Nevertheless, your question prompted us to include an explicit strategy for choosing $\alpha$, which enhances the completeness and clarity of our work. Thank you!

---

> ### Author Response · Authors · 2025-11-22
> **Response to Reviewer ivSN (two)**
>
> A1:
> Theorems 4.8: Suppose $f$ is in the general form described in problem 1 and assumption 2.1,2.2 hold. Suppose the update rule of $ x ^{t+1}$ follows Eq. 7. Assume the step size follows the decreasing form $ \eta_t = \frac{l}{\gamma + t}$, where $\gamma >0$ and the intermediate parameter $l=\frac{3}{\gamma_l\lambda_{\min}(\mathbf{M} )}$. Let $t_{\max} = T$ and $\alpha\le \sqrt{\frac{\alpha_0}{T+1}}$, where $\alpha_0$ is a tunable perturbation scale. We define $Q_2(\alpha_0^2)=\frac{3\left[6\gamma_u\lambda _{\max}(\mathbf{M})+36\gamma_u\gamma_l\mathrm{tr}(\mathbf{M} )\right]\lambda^2 _{\max}(\mathbf{M})(6+d)^3\gamma_u^2\alpha_0^2}{4\gamma_l^2\lambda^2 _{\min}(\mathbf{M} )}$. The initial step size satisfies $\eta _0=\frac{l}{\gamma }\le \frac{1}{12\gamma_u\mathrm{tr}(\mathbf{M})}$, which implies $\gamma  \geq \frac{36\gamma_u\mathrm{tr}(\mathbf{M})}{\lambda _{\min}(\mathbf{M})}$. Next, define the auxiliary parameter  $v  =\max\\{ \gamma (f( x ^0)-f^{\*}),\frac{54\gamma_u\mathrm{tr}(\mathbf{M})\sigma ^2}{\gamma_l^2\lambda^2 _{\min}(\mathbf{M})} +Q_2(\alpha_0^2) \\}$.
>
> We define $\mathrm{tr}(\mathbf{M})=\max_{  z ^t}\mathrm{tr}(\mathbf{M}(  z ^t))$, with similar definitions for $\lambda _{\min}(\mathbf{M})$ and $\lambda _{\max}(\mathbf{M})$. Then, for every integer $t$ with $0 \le t \le T$, we can obtain the following result
> \begin{align*}
> 	\mathbb{E}\left[f( x ^{t})-f^{\*}\right]\leq \frac{v  }{\gamma +t}.
> \end{align*}
>
> Q2:Failure to handle stochastic functions. Because the variance term grows with T, the analysis effectively applies only to deterministic or finite-sum settings.
> There is no uniform bound on stochastic noise, so the claimed results do not establish convergence for genuinely stochastic oracles. The theory should have been presented as deterministic analysis rather than stochastic convergence. How could the variance term be controlled to extend the results to true stochastic settings? Maybe use momentum?
>
> A2:We have revised the statements of Theorems 4.5 and 4.8 in A1. These modifications do not affect our final conclusions or contributions in the stochastic settings.
>
> Q3:Lack of transparency. By labeling ($Q_1$,$Q_2$) as constants and not clarifying their T-dependence, the manuscript obscures a fundamental limitation of the analysis. Can you restate Theorems 4.5 and 4.8 with explicit T-dependence and honest asymptotic interpretation?
>
> A3:To address potential ambiguities or misunderstandings, we have revised the statements of Theorems 4.5 and 4.8 in A1.
>
> Q4:Ambiguity in Assumption 2.1. The constants ($\gamma_u$,$\gamma_l$) are written as if they may depend on x,y,z, which would make the inequalities tautological.
> For the theorems to hold, they must be global constants independent of those points.
> This appears to be a typographical error that needs correction.
>
> A4:Thank you for the suggestion. We have revised Assumption 2.1 to eliminate potential ambiguities. $\gamma_u$ and $\gamma_l$ are all global constants.
>
> Q5:Lack of discussion of when assumptions hold. The paper should explicitly identify and justify classes of functions satisfying QR (e.g., quadratics, certain regularized GLMs). Beyond the trivial quadratic case, examples are only hinted at and never proved. Can you formally characterize non-quadratic functions (e.g., logistic or least-squares objectives) that satisfy the QR condition with global constants?
>
> A5:We agree that it is important to clarify when the quadratic regularity (QR) assumption holds. Our QR assumption is directly inherited from Frangella et al. (2024), where it is shown that QR is satisfied under many standard conditions. Frangella et al. (2024) provide two sufficient conditions under which the quadratic regularity assumption holds:
>
> 1.If the function $f$ is $L$-smooth and $\mu$-strongly convex, then $f$ satisfies the quadratic regularity assumption.
>
> 2.If the function $f$ is $\mu$-strongly convex and has a $\mathbf{M}$-Lipschitz Hessian, then $f$ satisfies the quadratic regularity assumption.
>
> From the above sufficient conditions, we can conclude that both the least-squares objectives and the regularized logistic regression objectives used in our experiments satisfy the quadratic regularity assumption. Due to space limitations, we plan to include these discussions in the camera-ready version.

---

> ### Author Response · Authors · 2025-11-22
> **Response to Reviewer ivSN (three)**
>
> Q6:Limited novelty and scope. Algorithmically, ZSG is standard (Gaussian SPSA / NES).
> Experiments are small-scale and deterministic; no tests on high-variance or nonconvex settings.
>
> A6:First, we would like to clarify that our contribution does not lie in proposing a new algorithm, but rather in uncovering the underlying reason why ZSG often outperforms ZSC in practice. This theoretical insight is currently missing in the academic community, and our work provides a novel contribution in this regard. Second, extensive experiments are not the main focus of this paper, our primary objective is to analyze the superiority of ZSG from a theoretical perspective. There already exist numerous works on fine-tuning LLMs that provide ample practical guidance and empirical evidence, such as (Malladi et al., 2023), (Zhao et al., 2024), (Zhang et al., 2024), (Chen et al., 2024), and so on. Finally, we theoretically validate the superiority of ZSG in the non-convex settings. We provide a simplified version of the analysis below. For clarity, we temporarily ignore the term $\phi( u , \alpha,  x )$, omit the quadratic lower-bound inequality, and make an initial assumption that $\eta_t\le \frac{1}{3\gamma_u\mathrm{tr}(\mathbf{M})}$. The main steps of ZSG are as follows:
> \begin{align*}
> 	f( x ^{t+1})
> 	\le&
> 	f( x ^t) -\eta_t\left\langle \nabla f( x ^t),  u _t u _t^\top\nabla f( x ^t,\mathcal{S}_t)\right\rangle +\frac{\gamma_u\eta_t^2}{2}\lVert  u _t u _t^\top\nabla f( x ^t,\mathcal{S}_t)\rVert _{\mathbf{M}(  z ^t) }^2.
> \end{align*}
>  \begin{align*}
> 	\mathbb{E}\left[f( x ^{t+1})\right]
> 	\le& f( x ^t)-\eta_t\mathbb{E}\left[\lVert \nabla f( x ^t)\rVert ^2\right]+\frac{3\gamma_u\eta_t^2\mathrm{tr}(\mathbf{M})}{2}\mathbb{E}\left[\lVert \nabla f( x ^t)\rVert ^2\right]+\frac{3\gamma_u\eta_t^2\mathrm{tr}(\mathbf{M})\sigma^2}{2}.
> \end{align*}
>  \begin{align*}
> 	\mathbb{E}\left[f( x ^{t+1})\right]
> 	\le& f( x ^t)-\eta_t(1-\frac{3\gamma_u\mathrm{tr}(\mathbf{M})}{2}\eta_t)\mathbb{E}\left[\lVert \nabla f( x ^t)\rVert ^2\right]+\frac{3\gamma_u\eta_t^2\mathrm{tr}(\mathbf{M})\sigma^2}{2}.
> \end{align*}
>  \begin{align*}
> 	\mathbb{E}\left[f( x ^{t+1})\right]
> 	\le&f( x ^t)-\frac{\eta_t}{2}\mathbb{E}\left[\lVert \nabla f( x ^t)\rVert ^2\right]+\frac{3\gamma_u\eta_t^2\mathrm{tr}(\mathbf{M})\sigma^2}{2}.
> \end{align*}
> Transpose the terms to get
>  \begin{align*}
>       \frac{\eta_t}{2}\mathbb{E}\left[\lVert \nabla f( x ^t)\rVert^2\right]
>       \le
>       \mathbb{E}\left[f( x ^t)-f( x ^{t+1})\right]
>       + \frac{3\gamma_t \mathrm{tr}(\mathbf{M})\sigma^2 \eta_t^2}{2}.
> \end{align*}
>
>  \begin{align*}
>       \frac{\sum_{t=0}^{T-1} \eta_t\mathbb{E}\left[\lVert\nabla f(x^t)\rVert^2\right]}{2}
>       \le
>       \sum_{t=0}^{T-1} \mathbb{E}\left[f( x ^t)-f( x ^{t+1})\right] + \frac{3\gamma_u \mathrm{tr}(\mathbf{M})\sigma^2\sum_{t=0}^{T-
>      1}\eta_t^2}{2}.
> \end{align*}
>
>  \begin{align*}
>       \frac{\sum_{t=0}^{T-1} \eta_t\mathbb{E}\left[\lVert\nabla f(x^t)\rVert^2\right]}{2}
>       \le
>       (f( x ^0)-f^{\*}) + \frac{3\gamma_u \mathrm{tr}(\mathbf{M})\sigma^2\sum_{t=0}^{T-1}\eta_t^2}{2}.
> \end{align*}
> So, we can obtain
>
> \begin{align*}
>     \min_{0\le t \le T-1}
>     \mathbb{E}\left[\lVert\nabla f( x ^t)\rVert^2\right]
>     \le
>     \frac{2(f(x^0)-f^{\*}) + 3\gamma_u\mathrm{tr}(\mathbf{M})\sigma^2 \sum_{t=0}^{T-1} \eta_t^2}{\sum_{t=0}^{T-1} \eta_t}.
> \end{align*}
> Furthermore, if we choose $\eta_t=\eta=\min \\{ \frac{1}{3 \gamma_u \mathrm{tr}(\mathbf{M})},\sqrt{\frac{2(f( x ^0)-f^{\*})}{3 \gamma_u \mathrm{tr}(\mathbf{M})\sigma^2 T}} \\}$, we can obtain
> \begin{align*}
>     \min_{0\le t \le T-1}
>     \mathbb{E}\left[\lVert\nabla f( x ^t)\rVert^2\right]=\mathcal{O}(\sqrt{\frac{\gamma_u \mathrm{tr}(\mathbf{M})\sigma^2 (f( x ^0)-f^{\*})}{T}}).
> \end{align*}
> By the same reasoning, a convergence analysis can also be established for ZSC:
> \begin{align*}
>     \min_{0\le t \le T-1}
>     \mathbb{E}\left[\lVert\nabla f( x ^t)\rVert^2\right]
>     =\mathcal{O}(\sqrt{\frac{\gamma_u d\lambda _{\max}(\mathbf{M})\sigma^2 (f( x ^0)-f^{\*})}{T}}).
> \end{align*}
> Therefore, our theoretical insights continue to hold in the non-convex setting. Due to space limitations, we are unable to include the full analysis in the current version of the paper. If deemed necessary, we can incorporate a discussion of the non-convex case in the camera-ready version.

---

> ### Author Response · Authors · 2025-11-22
> **Response to Reviewer ivSN (four)**
>
> Q7:Would a fully deterministic framing ($\sigma = 0$) strengthen the paper?
>
> A7:First, we have revised the statements of Theorems 4.5 and 4.8 in A1. We establish the theoretical advantage of ZSG over ZSC in the stochastic settings. Second, the purpose of establishing the $\mathcal{O}\left(\frac{\mathrm{tr}(\mathbf{M})}{\lambda _{\min}(\mathbf{M})}\log\frac{1}{\varepsilon}\right)$ iteration complexity in the deterministic settings in Corollary 4.4 is to demonstrate that an intermediate result of our analysis subsumes the theoretical contribution of Wang et al. (2024). This demonstrates that our work is more comprehensive and profound.
>
> Reference List
>
> [1]Sadhika Malladi, Tianyu Gao, Eshaan Nichani, Alex Damian, Jason D Lee, Danqi Chen, and Sanjeev Arora. Fine-tuning language models with just forward passes. Advances in Neural Information Processing Systems, 36:53038–53075, 2023.
>
> [2]Yanjun Zhao, Sizhe Dang, Haishan Ye, Guang Dai, Yi Qian, and Ivor W Tsang. Second-order fine-tuning without pain for llms: A hessian informed zeroth-order optimizer. arXiv preprint arXiv:2402.15173, 2024.
>
> [3]J.C. Spall. Multivariate stochastic approximation using a simultaneous perturbation gradient approximation. IEEE Transactions on Automatic Control, 37(3):332–341, 1992. doi: 10.1109/9.119632.
>
> [4]Yurii Nesterov and Vladimir Spokoiny. Random gradient-free minimization of convex functions. Foundations of Computational Mathematics, 17(2):527–566, 2017.
>
> [5]B.Polyak. Introduction to Optimization. Optimization Software - Inc., Publications
> Division, New York (1987).
>
> [6]Zachary Frangella, Pratik Rathore, Shipu Zhao, and Madeleine Udell. Promise: Preconditioned stochastic optimization methods by incorporating scalable curvature estimates. arXiv preprint arXiv:2309.02014, 2023.
>
> [7]Zhang, Yihua, Pingzhi Li, Junyuan Hong, Jiaxiang Li, Yimeng Zhang, Wenqing Zheng, Pin-Yu Chen et al. "Revisiting zeroth-order optimization for memory-efficient llm fine-tuning: A benchmark." arXiv preprint arXiv:2402.11592 (2024).
>
> [8]Chen, Yiming, Yuan Zhang, Liyuan Cao, Kun Yuan, and Zaiwen Wen. "Enhancing zeroth-order fine-tuning for language models with low-rank structures." arXiv preprint arXiv:2410.07698 (2024).
>
> [9]Yilong Wang, Haishan Ye, Guang Dai, and Ivor Tsang. Can gaussian sketching converge faster on a preconditioned landscape? In Forty-first International Conference on Machine Learning, 2024.

---

### Official Review · Reviewer_HDZg · 2025-10-31

**Soundness:** 3
**Presentation:** 2
**Contribution:** 3
**Rating:** 6
**Confidence:** 2

**Summary:**

This paper establishes an accelerated convergence rate for ZSG and theoretically analyze the potential advantages of ZSG compared to ZSC. The paper evaluates on both synthetic and real-world datasets and demonstrate the performance of ZSG outperforms that of ZSC.

**Strengths:**

Novel analysis:
Novel theoretical contribution by being the first to rigorously analyze why ZSG outperforms ZSC in practice

The theoretical gap addressed is important:
The work addresses why ZSG is preferred in practice despite identical O(d) complexity bounds

Clarity:
The paper is generally well written with clear problem setup and algorithmic description. The contributions and stated clearly and the main results are stated precisely with appropriate assumptions.

**Weaknesses:**

Notations:
There is heavy notation that accumulates through the paper that can make it difficult to parse.


Assumptions do not match motivating examples:
All results assume strongly convex objectives (Assumption 2.1), but the examples used to motivate the analysis such as LLM fine tuning involve non-convex deep learning problems. This means the analysis cannot be directly applied to the examples it states


Experiments:
Some experimental details are sparse such as how are step sizes chosen. The experimental evaluations are very toy settings. However this may be fine because the work mainly fills a theoretical gap.

Missing comparisons:
The paper does not provide theoretical or empirical comparison to variance reduced zeroth-order methods or adaptive/momentum based zeroth order methods.

**Questions:**

How should users set the step size when $tr(M)$, $\lambda_\min(M)$ are unknown?

Is there a way to construct an experiment for LLM fine tuning and other motivating examples?

---

> ### Author Response · Authors · 2025-11-22
> **Response to Reviewer HDZg (one)**
>
> We sincerely appreciate your insightful comments. We will address each of the concerns and issues you raised.
>
> Q1:Assumptions do not match motivating examples. All results assume strongly convex objectives (Assumption 2.1), but the examples used to motivate the analysis such as LLM fine tuning involve non-convex deep learning problems. This means the analysis cannot be directly applied to the examples it states.
>
> A1:Extending the convergence theorem of ZSG to the non-convex setting is a straightforward extension of our analysis. We provide a simplified version of the analysis below. For clarity, we temporarily ignore the term $\phi( u , \alpha,  x )$, omit the quadratic lower-bound inequality, and make an initial assumption that $\eta_t\le \frac{1}{3\gamma_u\mathrm{tr}(\mathbf{M})}$.
> Here, $\mathbf{M}$ is no longer required to be the Hessian matrix, rather, this follows the standard $\mathbf{M}$-smoothness assumption ($L$–smoothness assumption is a special case of $\mathbf{M}$–smoothness for $\gamma_u\mathbf{M}=L\mathbf{I}$). The main steps of ZSG are as follows:
> \begin{align*}
> 	f( x ^{t+1})
> 	\le&
> 	f( x ^t) -\eta_t\left\langle \nabla f( x ^t),  u _t u _t^\top\nabla f( x ^t,\mathcal{S}_t)\right\rangle +\frac{\gamma_u\eta_t^2}{2}\lVert  u _t u _t^\top\nabla f( x ^t,\mathcal{S}_t)\rVert _{\mathbf{M}}^2.
> \end{align*}
>  \begin{align*}
> 	\mathbb{E}\left[f( x ^{t+1})\right]
> 	\le& f( x ^t)-\eta_t\mathbb{E}\left[\lVert \nabla f( x ^t)\rVert ^2\right]+\frac{3\gamma_u\eta_t^2\mathrm{tr}(\mathbf{M})}{2}\mathbb{E}\left[\lVert \nabla f( x ^t)\rVert ^2\right]+\frac{3\gamma_u\eta_t^2\mathrm{tr}(\mathbf{M})\sigma^2}{2}.
> \end{align*}
>  \begin{align*}
> 	\mathbb{E}\left[f( x ^{t+1})\right]
> 	\le& f( x ^t)-\eta_t(1-\frac{3\gamma_u\mathrm{tr}(\mathbf{M})}{2}\eta_t)\mathbb{E}\left[\lVert \nabla f( x ^t)\rVert ^2\right]+\frac{3\gamma_u\eta_t^2\mathrm{tr}(\mathbf{M})\sigma^2}{2}.
> \end{align*}
>  \begin{align*}
> 	\mathbb{E}\left[f( x ^{t+1})\right]
> 	\le&f( x ^t)-\frac{\eta_t}{2}\mathbb{E}\left[\lVert \nabla f( x ^t)\rVert ^2\right]+\frac{3\gamma_u\eta_t^2\mathrm{tr}(\mathbf{M})\sigma^2}{2}.
> \end{align*}
> Transpose the terms to get
>  \begin{align*}
>       \frac{\eta_t}{2}\mathbb{E}\left[\lVert \nabla f( x ^t)\rVert^2\right]
>       \le
>       \mathbb{E}\left[f( x ^t)-f( x ^{t+1})\right]
>       + \frac{3\gamma_t \mathrm{tr}(\mathbf{M})\sigma^2 \eta_t^2}{2}.
> \end{align*}
>
>  \begin{align*}
>       \frac{\sum_{t=0}^{T-1} \eta_t\mathbb{E}\left[\lVert\nabla f(x^t)\rVert^2\right]}{2}
>       \le
>       \sum_{t=0}^{T-1} \mathbb{E}\left[f( x ^t)-f( x ^{t+1})\right] + \frac{3\gamma_u \mathrm{tr}(\mathbf{M})\sigma^2\sum_{t=0}^{T-
>      1}\eta_t^2}{2}.
> \end{align*}
>
>  \begin{align*}
>       \frac{\sum_{t=0}^{T-1} \eta_t\mathbb{E}\left[\lVert\nabla f(x^t)\rVert^2\right]}{2}
>       \le
>       (f( x ^0)-f^{\*}) + \frac{3\gamma_u \mathrm{tr}(\mathbf{M})\sigma^2\sum_{t=0}^{T-1}\eta_t^2}{2}.
> \end{align*}
> So, we can obtain
>
> \begin{align*}
>     \min_{0\le t \le T-1}
>     \mathbb{E}\left[\lVert\nabla f( x ^t)\rVert^2\right]
>     \le
>     \frac{2(f(x^0)-f^{\*}) + 3\gamma_u\mathrm{tr}(\mathbf{M})\sigma^2 \sum_{t=0}^{T-1} \eta_t^2}{\sum_{t=0}^{T-1} \eta_t}.
> \end{align*}
> Furthermore, if we choose $\eta_t=\eta=\min \\{ \frac{1}{3 \gamma_u \mathrm{tr}(\mathbf{M})},\sqrt{\frac{2(f( x ^0)-f^{\*})}{3 \gamma_u \mathrm{tr}(\mathbf{M})\sigma^2 T}} \\}$, we can obtain
> \begin{align*}
>     \min_{0\le t \le T-1}
>     \mathbb{E}\left[\lVert\nabla f( x ^t)\rVert^2\right]=\mathcal{O}(\sqrt{\frac{\gamma_u \mathrm{tr}(\mathbf{M})\sigma^2 (f( x ^0)-f^{\*})}{T}}).
> \end{align*}
> By the same reasoning, a convergence analysis can also be established for ZSC:
> \begin{align*}
>     \min_{0\le t \le T-1}
>     \mathbb{E}\left[\lVert\nabla f( x ^t)\rVert^2\right]
>     =\mathcal{O}(\sqrt{\frac{\gamma_u d\lambda _{\max}(\mathbf{M})\sigma^2 (f( x ^0)-f^{\*})}{T}}).
> \end{align*}
> Therefore, our theoretical insights continue to hold in the non-convex setting. Due to space limitations, we are unable to include the full analysis in the current version of the paper. If deemed necessary, we can incorporate a discussion of the non-convex case in the camera-ready version.

---

> ### Author Response · Authors · 2025-11-22
> **Response to Reviewer HDZg (two)**
>
> Q2:Some experimental details are sparse such as how are step sizes chosen. The experimental evaluations are very toy settings. However this may be fine because the work mainly fills a theoretical gap. And how should users set the step size when
> $\mathrm{tr}(\mathbf{M})$, $\lambda _{\min}(\mathbf{M})$ are unknown?
>
> A2:We are glad that you recognize our main contribution, which lies in filling this theoretical gap and uncovering the fundamental reason why ZSG often outperforms ZSC in practice. When information related to the step-size is difficult to obtain in practice, a common strategy is to start with a relatively small step-size and then adjust it progressively. Rakhlin et al. (2011) have established the optimal convergence rate $\mathbb{E}\!\left[F(\mathbf{w}_T)-F(\mathbf{w}^*)\right] \le \frac{2\mu G^2}{\lambda^2 T}$, along with the recommended step-size $\eta_t = \frac{1}{\lambda t}$ for SGD, where $\lambda$ denotes the strong convexity parameter. Their work also focuses on the theoretical aspects rather than practical step-size tuning strategies. Malladi et al. (2023) and Zhao et al. (2024) also employ zeroth-order optimization methods for fine-tuning LLMs and attempt to provide convergence analyses. Similarly, they rely on theoretically appropriate step-sizes $\eta_t$ when deriving their convergence guarantees, while in practice they simply select several small, commonly used learning rates for the experiments. In summary, we consider that the discussion of step-size selection strategies is not central to the main scope of our work.
>
> Q3:Missing comparisons. The paper does not provide theoretical or empirical comparison to variance reduced zeroth-order methods or adaptive/momentum based zeroth order methods.
>
> A3:We appreciate the suggestions regarding variance reduced or adaptive/momentum algorithms. We would like to emphasize, however, that extending poses some technical challenges. For this reason, our current contribution focuses on SGD rather than variance reduction methods. These extensions can serve as potential directions for our future work. We also believe our findings have the potential to inspire future efforts within the optimization and machine learning communities to tackle this open problem.
>
> Q4:Is there a way to construct an experiment for LLM fine tuning and other motivating examples?
>
> A4:The main contribution of our work lies in providing a theoretical explanation for why researchers tend to prefer ZSG over ZSC in real-world applications. Numerous studies have already employed ZSG for fine-tuning large language models, while ZSC is typically abandoned in practice such as (Malladi et al., 2023), (Zhao et al., 2024), (Zhang et al., 2024), (Chen et al., 2024), and so on. Therefore, we believe that reproducing these experiments would be unnecessary.
>
> Reference List
>
> [1]Alexander Rakhlin, Ohad Shamir, and Karthik Sridharan. Making gradient descent optimal for strongly convex stochastic optimization. arXiv preprint arXiv:1109.5647, 2011.
>
> [2]Sadhika Malladi, Tianyu Gao, Eshaan Nichani, Alex Damian, Jason D Lee, Danqi Chen, and Sanjeev Arora. Fine-tuning language models with just forward passes. Advances in Neural Information Processing Systems, 36:53038–53075, 2023.
>
> [3]Yanjun Zhao, Sizhe Dang, Haishan Ye, Guang Dai, Yi Qian, and Ivor W Tsang. Second-order fine-tuning without pain for llms: A hessian informed zeroth-order optimizer. arXiv preprint arXiv:2402.15173, 2024.
>
> [4]Zhang, Yihua, Pingzhi Li, Junyuan Hong, Jiaxiang Li, Yimeng Zhang, Wenqing Zheng, Pin-Yu Chen et al. "Revisiting zeroth-order optimization for memory-efficient llm fine-tuning: A benchmark." arXiv preprint arXiv:2402.11592 (2024).
>
> [5]Chen, Yiming, Yuan Zhang, Liyuan Cao, Kun Yuan, and Zaiwen Wen. "Enhancing zeroth-order fine-tuning for language models with low-rank structures." arXiv preprint arXiv:2410.07698 (2024).

---

### Official Review · Reviewer_hEb7 · 2025-11-01

**Soundness:** 3
**Presentation:** 3
**Contribution:** 2
**Rating:** 6
**Confidence:** 2

**Summary:**

The paper studies stochastic zeroth-order optimization and compares Gaussian perturbation ZO-SGD (ZSG) with coordinate finite-difference ZO-SGD (ZSC). Under a quadratic regularity assumption that lifts smooth/strong-convex conditions to a Hessian-norm form, it proves that ZSG enjoys weaker dimension dependence and faster convergence than ZSC, especially under skewed Hessian spectra.

**Strengths:**

This paper is easy to navigate: assumptions and notation are stated upfront, the notion of quadratic regularity is introduced with intuition before formal use.  And the Theorems show ZSG attains iteration/query bounds that avoid the explicit factor 𝑑 that appears for ZSC.

**Weaknesses:**

**Question 1.** The bounds hinge on $\gamma_u, \gamma_l$ and on quantities like $\operatorname{tr}(M), \lambda_{\min }(M), \lambda_{\max }(M)$, which may be unknown or hard to estimate. Thus, practical guidance for choosing $\eta_t$ that depends on these is limited.

**Question 2.** The empirical comparison is primarily ZSG vs. ZSC; other ZO baselines (e.g., two-point random directions with mini-batching/importance sampling) are not reported, making it harder to gauge practical significance.

**Question 3.**  Although the theory targets skewed spectra, experiments do not directly measure Hessian anisotropy on real data (only synthetic constructions), so the claimed mechanism is not empirically verified on those tasks.

**Question 4.** Sensitivity to $\alpha$ and noise assumptions. Theory requires "sufficiently small" $\alpha$ and a bounded variance $\sigma^2$; experiments fix $\alpha=10^{-6}$ without sensitivity analysis, so robustness is unclear.

**Questions:**

See the weakness.

---

> ### Author Response · Authors · 2025-11-22
> **Response to Reviewer hEb7**
>
> Thank you very much for your feedback. We will carefully address each of the concerns and issues you raised.
>
> Q1:The bounds hinge on $\gamma_u$, $\gamma_l$ and on quantities like $\mathrm{tr}(\mathbf{M})$, $\lambda _{\min}(\mathbf{M})$, $\lambda _{\max}(\mathbf{M})$
> , which may be unknown or hard to estimate. Thus, practical guidance for choosing $\eta _t$ that depends on these is limited.
>
> A1:Your concern is valid and we will clarify this point. While the bounds we derive do depend on quantities that may not be known precisely, our intent is to provide a theoretical explanation for the superiority of ZSG. The exact values of these quantities are not required for our theoretical conclusions to hold. As is common in the analysis of SGD-type methods, the step-size used in theory primarily serves to facilitate convergence analysis. In practice, one typically starts with a relatively small step-size and then tunes it progressively. Considering the objective function $F$ is both strongly convex and smooth, Rakhlin et al. (2011) have established the optimal convergence rate $\mathbb{E}\!\left[F(\mathbf{w}_T)-F(\mathbf{w}^*)\right] \le \frac{2\mu G^2}{\lambda^2 T}$, along with the recommended step-size $\eta_t = \frac{1}{\lambda t}$ for SGD, where $\lambda$ denotes the strong convexity parameter. Their work also focuses on the theoretical aspects rather than practical step-size tuning strategies. Malladi et al. (2023) and Zhao et al. (2024) also employ zeroth-order optimization methods for fine-tuning LLMs and attempt to provide convergence analyses. Similarly, they rely on theoretically appropriate step-sizes $\eta_t$ when deriving their convergence guarantees, while in practice they simply select several small, commonly used learning rates for the experiments. In summary, the convergence analysis primarily serves as a theoretical contribution, whereas the choice and tuning of learning-rate schedules are engineering considerations. Our work focuses on addressing the theoretical gap underlying the superior performance of ZSG.
>
> Q2:The empirical comparison is primarily ZSG vs. ZSC; other ZO baselines (e.g., two-point random directions with mini-batching/importance sampling) are not reported, making it harder to gauge practical significance.
>
> A2:ZSG and ZSC are both based on two-point gradient estimators and are further combined with mini-batch sampling techniques. In line 147 of the paper, we define ''$\nabla f( x ,\mathcal{S})=\frac{1}{|\mathcal{S}|}\sum_{j\in \mathcal{S}}^{}\nabla f_j( x )$, where $\mathcal{S}$ represents the sample set and $|\mathcal{S}|$ represents the sample size.'' The corresponding description in Algorithm 1 also supports this point. Second, zeroth-order optimization methods access the objective function only in a black-box manner. Since gradient information is not available, it is not possible to perform importance sampling.
>
> Q3:Although the theory targets skewed spectra, experiments do not directly measure Hessian anisotropy on real data (only synthetic constructions), so the claimed mechanism is not empirically verified on those tasks.
>
> A3:We have followed your suggestion and plotted the eigenvalue distribution of the Hessian matrix on real-data experiments in Section F. We find that all of these datasets fall into the skewed Hessian setting. We sincerely appreciate your constructive feedback, which has helped improve the quality of our paper.
>
> Q4:Sensitivity to $\alpha$ and noise assumptions. Theory requires "sufficiently small" $\alpha$ and a bounded variance $\sigma^2$; experiments fix $\alpha=10^{-6}$ without sensitivity analysis, so robustness is unclear.
>
> A4:In the theoretical analysis of SGD, variance control is typically treated as a standard assumption. Furthermore, we agree that conducting a sensitivity analysis with respect to $\alpha$ is valuable. We have added the corresponding experiments in Section G. We appreciate your helpful suggestion. We find that an  large $\alpha$ significantly hinders the convergence process, while $\alpha = 10^{-2}$ is already sufficient for most cases.
>
> Reference List
>
> [1]Alexander Rakhlin, Ohad Shamir, and Karthik Sridharan. Making gradient descent optimal for strongly convex stochastic optimization. arXiv preprint arXiv:1109.5647, 2011.
>
> [2]Sadhika Malladi, Tianyu Gao, Eshaan Nichani, Alex Damian, Jason D Lee, Danqi Chen, and Sanjeev Arora. Fine-tuning language models with just forward passes. Advances in Neural Information Processing Systems, 36:53038–53075, 2023.
>
> [3]Yanjun Zhao, Sizhe Dang, Haishan Ye, Guang Dai, Yi Qian, and Ivor W Tsang. Second-order fine-tuning without pain for llms: A hessian informed zeroth-order optimizer. arXiv preprint arXiv:2402.15173, 2024.

---

### Author Response · Authors · 2025-11-22
**To Reviewers**

We sincerely appreciate the reviewers’ constructive comments and suggestions. We have carefully revised the manuscript accordingly and have now submitted the updated version for you.

---

### Meta-Review · Area_Chair_hEWw · 2025-12-30

**Summary:**

The paper studies zeroth-order stochastic optimization for large-scale finite-sum problems, with a focus on comparing Gaussian-based and coordinate-wise gradient estimators. It introduces a quadratic regularity assumption that extends standard smoothness and strong convexity conditions and enables the use of Hessian-related information in the convergence analysis. Within this framework, the authors derive convergence guarantees suggesting an advantage for Gaussian-based zeroth-order SGD over coordinate-based methods. Experimental results on synthetic and real-world datasets are provided to support the theoretical analysis.

**Reviewer Concerns:**

Below, I summarize the main concerns raised by the reviewers and also explain how the authors addressed the criticism.

**Reviewer hEb7.**

1. The convergence bounds depend on Hessian-related quantities that may be difficult to estimate in practice, limiting actionable guidance. The authors clarify that Hessian-dependent quantities are used for theoretical insight rather than practical tuning, and argue that step-size choices serve analysis purposes, as is standard in SGD theory.

2. Experimental comparisons are limited to coordinate-wise ZO-SGD, omitting other standard zeroth-order baselines. The authors argue that ZSG and ZSC already represent comparable two-point, mini-batch zeroth-order methods, and that other baselines (e.g., importance sampling) are not applicable in black-box zeroth-order settings.

3. The proposed advantage under skewed Hessian spectra is not directly validated on real-world datasets. In response to the concern about Hessian anisotropy, the authors add new experiments that explicitly plot Hessian eigenvalue distributions on real datasets, showing skewed spectra consistent with their theory.

4. Sensitivity to smoothing and noise assumptions is not examined, leaving robustness unclear. The authors acknowledge the lack of sensitivity analysis for the smoothing parameter and add new experiments demonstrating that overly large values degrade performance, while the chosen small value is sufficient in practice.

**Reviewer HDZg.**

1. The paper relies on strong convexity assumptions, while some motivating examples (e.g., LLM fine-tuning) are non-convex, limiting the direct applicability of the theoretical results. The authors argue that their theoretical insights extend to the non-convex setting and provide a sketch of a convergence analysis for ZSG (and ZSC) under standard smoothness assumptions.

2. Heavy and accumulating notation makes parts of the paper difficult to parse. The authors didn't address this one, but I think it is relatively minor.

3. Experimental details are limited (e.g., step-size selection), and the evaluations are relatively toy, though this may be acceptable given the primarily theoretical focus. The authors emphasize that the paper’s primary contribution is theoretical, and that practical step-size selection follows common heuristic practices (e.g., starting small and tuning), consistent with prior SGD and zeroth-order literature.

4. The paper lacks theoretical or empirical comparisons with variance-reduced, adaptive, or momentum-based zeroth-order methods. The authors state that extending the analysis to variance-reduced or adaptive/momentum-based zeroth-order methods poses technical challenges and is left as future work.

**Reviewer ivSN.**

1. The main convergence theorems are mis-stated, as key constants depend on the total iteration count, so the bounds do not imply vanishing error or true stochastic convergence.

2. Due to variance terms growing with iterations, the analysis effectively applies only to deterministic or finite-sum settings rather than genuinely stochastic oracles.

3. Assumption 2.1 (quadratic regularity) is ambiguously defined, with constants potentially depending on problem instances, which would invalidate the stated inequalities.

4. The paper lacks a clear characterization and justification of when the quadratic regularity assumption holds beyond trivial quadratic cases.

5. The novelty and scope are limited, as the algorithmic setting is standard and experiments are small-scale and deterministic, with no evaluation in high-variance or non-convex regimes.

6. The presentation obscures limitations by labeling iteration-dependent quantities as constants, reducing transparency about the true strength of the results.

The authors revised the main theorems to clarify iteration and smoothing-parameter dependence, fixed ambiguities in the quadratic regularity assumption, and explained that convergence relies on taking the perturbation parameter sufficiently small, emphasizing that the contribution is primarily theoretical and explanatory. They also discussed when the quadratic regularity assumption holds and sketched extensions to non-convex settings. The reviewer acknowledged that these clarifications address the core issue regarding bias and parameter scaling, but noted that these points should have been made clearly from the outset. What remains missing is a fully developed stochastic convergence theory with uniform noise control, inclusion of the non-convex analysis within the paper itself rather than in rebuttal, and a clearer upfront positioning of the results as conditional and asymptotic.

**Reviewer XZDu.**

1. The claimed negligibility of higher-order error terms with small smoothing parameter is questionable, as these terms include large dimension- and Hessian-dependent factors. The authors clarified that their analysis explicitly retains the full bias term and relies on standard zeroth-order assumptions where the smoothing parameter is taken to zero, revising the theorems to make this dependence explicit and asymptotically valid.

2. The presentation is unclear, with missing or insufficient descriptions of baseline methods (e.g., ZSC) and difficulty distinguishing between closely related theorems. The authors explained that ZSC differs from ZSG only in the choice of random directions, added an explicit description and flowchart in the appendix, and clarified the roles of the closely related theorems.

3. The use of worst-case Hessian trace and eigenvalue bounds is overly coarse and may blur distinctions between merely convex and strongly convex problems. The authors clarified that strong convexity is explicitly assumed where needed, justified the use of worst-case trace and eigenvalue bounds, and provided additional discussion and analysis extending the framework beyond purely quadratic objectives.

4. The experimental evaluation is limited, lacking confidence intervals and more challenging or realistic benchmarks beyond small-scale datasets. The authors emphasized that the paper’s primary contribution is theoretical, argued that existing experiments are sufficient to validate the theory, and noted that ZSG is already widely adopted in practice, while acknowledging that additional empirical studies could be beneficial but are outside scope.

**Reviewer Scores:**

**Reviewer hEb7.** The authors appear to have adequately addressed the reviewer’s concerns. However, given the reviewer’s low confidence score, it is unlikely that these clarifications would lead to a change in the final recommendation.

**Reviewer HDZg.** The authors have largely addressed the raised issues. That said, the reviewer’s low confidence score suggests that a revision of the rating is improbable.

**Reviewer ivSN.** As discussed above, the authors partially addressed the reviewer’s concerns through clarifications and revisions. Since the reviewer did not revise their score after the initial responses, it is unlikely that the score would increase following the rebuttal.

**Reviewer XZDu.** It is unclear whether the authors’ responses would be sufficient to change the reviewer’s assessment.

**Overall assessment.** While the paper contains interesting ideas and potentially valuable theoretical insights, the rebuttal indicates that substantial modifications and clarifications are still required, and several concerns have only been partially addressed. In particular, key aspects of the theoretical scope, assumptions, and presentation remain insufficiently resolved. The paper would benefit from an additional full review cycle before a final decision can be made. Therefore, I recommend rejection at this stage.

---

### Decision · Program_Chairs · 2026-01-26

Reject